# INCOMPATIBILITY CLUSTERING AS A DEFENSE AGAINST BACKDOOR POISONING ATTACKS

**Charles Jin**
CSAIL
MIT
Cambridge, MA 02139
ccj@csail.mit.edu

**Melinda Sun**
MIT
Cambridge, MA 02139
mmsun@mit.edu

**Martin Rinard**
CSAIL
MIT
Cambridge, MA 02139
rinard@csail.mit.edu

## ABSTRACT

We propose a novel clustering mechanism based on an *incompatibility property* between subsets of data that emerges during model training. This mechanism partitions the dataset into subsets that *generalize only to themselves*, i.e., training on one subset does not improve performance on the other subsets. Leveraging the interaction between the dataset and the training process, our clustering mechanism partitions datasets into clusters that are defined by—and therefore meaningful to—the objective of the training process.

We apply our clustering mechanism to defend against data poisoning attacks, in which the attacker injects malicious poisoned data into the training dataset to affect the trained model's output. Our evaluation focuses on backdoor attacks against deep neural networks trained to perform image classification using the GTSRB and CIFAR-10 datasets. Our results show that (1) these attacks produce poisoned datasets in which the poisoned and clean data are incompatible and (2) our technique successfully identifies (and removes) the poisoned data. In an end-to-end evaluation, our defense reduces the attack success rate to below 1% on 134 out of 165 scenarios, with only a 2% drop in clean accuracy on CIFAR-10 and a negligible drop in clean accuracy on GTSRB.

## 1 OVERVIEW

Clustering, which aims to partition a dataset to surface some underlying structure, has long been a fundamental technique in machine learning and data analysis, with applications such as outlier detection and data mining across diverse fields including personalized medicine, document analysis, and networked systems (Xu & Tian, 2015). However, traditional approaches scale poorly to large, high-dimensional datasets, which limits the utility of such techniques for the increasingly large, minimally curated datasets that have become the norm in deep learning (Zhou et al., 2022).

We present a new clustering technique that partitions the dataset according to an *incompatibility property* that emerges during model training. In particular, it partitions the dataset into subsets that *generalize only to themselves* (relative to the final trained model) and not to the other subsets. Directly leveraging the interaction between the dataset and the training process enables our technique to partition large, high-dimensional datasets commonly used to train neural networks (or, more generally, any class of learned models) into clusters defined by—and therefore meaningful to—the task the network is intended to perform.

Our technique operates as follows. Given a dataset, it iteratively samples a subset of the dataset, trains a model on the subset, then selects the elements within the larger dataset that the model scores most highly. These selected elements comprise a smaller dataset for the next refinement step. By decreasing the size of the selected subset in each iteration, this process produces a final subset that (ideally) contains only data which is compatible with itself. The identified subset is then removed from the starting dataset, and the process is repeated to obtain a collection of refined subsets and corresponding trained models. Finally, a majority vote using models trained on the resulting refined subsets merges compatible subsets to produce the final partitioning.

Our evaluation focuses on *backdoor data poisoning attacks* (Chen et al., 2017; Adi et al., 2018) against deep neural networks (DNNs), in which the attacker injects a small amount of poisoned data into the training dataset to install a backdoor that can be used to control the network's behavior during deployment. For example, Gu et al. (2019) install a backdoor in a traffic sign classifier, which causes the network to misclassify stop signs as speed limit signs when a (physical) sticker is applied. Prior work has found that directly applying classical outlier detection techniques to the dataset fails to separate the poisoned and clean data (Tran et al., 2018). A key insight is that training on poisoned data should not improve accuracy on clean data (and vice versa). Hence, the poisoned data satisfies the incompatibility property and can be separated using our clustering technique.

This paper makes the following contributions:

**Clustering with Incompatibility and Self-Expansion.** Section 2 defines *incompatibility* between subsets of data based on their interaction with the training algorithm, specifically that training on one subset does not improve performance on the other. We prove that for datasets containing incompatible subpopulations, the subset which is most compatible with itself (as measured by the *self-expansion error*) must be drawn from a single subpopulation. Hence, iteratively identifying such subsets based on this property provably separates the dataset along the original subpopulations. We present a tractable clustering mechanism that approximates this optimization objective.

**Formal Characterization and Defense for Data Poisoning Attacks.** Section 3 presents a formal characterization of data poisoning attacks based on incompatibility, namely, that the poisoned data is incompatible with the clean data. We provide theoretical evidence to support this characterization by showing that a backdoor attack against linear classifiers provably satisfies the incompatibility property. We present a defense that leverages the clustering mechanism to partition the dataset into incompatible clean and poisoned subsets.

**Experimental Evaluation.** Section 4 presents an empirical evaluation of the ability of the incompatibility property presented in Section 2 and the techniques developed in Section 3 to identify and remove poisoned data to defend against backdoor data poisoning attacks. We focus on three different backdoor attacks (two dirty label attacks using pixel-based patches and image-based patches, respectively, and a clean-label attack using adversarial perturbations) from the data poisoning literature (Gu et al., 2019; Gao et al., 2019; Turner et al., 2018). The results (1) indicate that the considered attacks produce poisoned datasets that satisfy the incompatibility property and (2) demonstrate the effectiveness of our clustering mechanism in identifying poisoned data within a larger dataset containing both clean and poisoned data. For attacks against the GTSRB and CIFAR-10 datasets, our defense mechanism reduces the attack success rate to below 1% on 134 out of the 165 scenarios, with a less than 2% drop in clean accuracy. Compared to three previous defenses, our method successfully defends against the standard dirty-label backdoor attack in 22 out of 24 scenarios, versus only 5 for the next best defense.

We have open sourced our implementation at `https://github.com/charlesjin/compatibility_clustering/`.

## 2 INCOMPATIBILITY CLUSTERING

This section presents our clustering technique for datasets with subpopulations of incompatible data. In our formulation, two subsets are incompatible when training on one subset does not improve performance on the other subset. Because we cannot measure performance on, e.g., holdout data with ground truth labels, we introduce a "self-expansion" property: given a set of data points, we first subsample the data to obtain a smaller training set, then measure how well the learning algorithm generalizes to the full set of data. Our main theoretical result is that a set which achieves *optimal* expansion is homogeneous, i.e., composed entirely of data from a single subpopulation. Finally, we propose the Inverse Self-Paced Learning algorithm, which uses an approximation of the self-expansion objective to partition the dataset.

### 2.1 SETTING

We present our theoretical results in the context of a basic binary classification setting. Let $X$ be the input space, $Y = \{-1, +1\}$ be the label space, and $L(\cdot, \cdot)$ be the 0-1 loss function over $Y \times Y$, i.e.,

$L(y_1, y_2) = (1 - y_1 * y_2)/2$. Given a target distribution $\mathcal{D}$ supported on $X \times Y$ and a parametric family of functions $f_\theta$, a standard objective in this setting is to find $\theta$ minimizing the population risk $R(\theta) := \int_{X \times Y} L(f_\theta(x), y) dP_{\mathcal{D}}(x, y)$. Given $n$ data points $D = \{(x_1, y_1), \ldots, (x_n, y_n)\}$, the empirical risk is defined as $R_{emp}(\theta; D) := n^{-1} \sum_{i=1}^{n} L(f_\theta(x_i), y_i)$. We fix a learning algorithm $\mathcal{A}(\cdot)$ which takes as input a training set and returns parameters $\theta$. For example, if $f_\theta$ is a family of DNNs, then $\mathcal{A}$ might train a DNN via stochastic gradient descent.

## 2.2 SELF-EXPANSION AND COMPATIBILITY

To measure generalization without holdout data and independent of any ground truth labels, we propose the following self-expansion property of sets:

**Definition 2.1** (Self-expansion of sets.). *Let $S$ and $T$ be sets ($S$ nonempty), and let $\alpha \in (0, 1]$ denote the* subsampling rate. *We define the $\alpha$-expansion error of $S$ given $T$ as*

$$\epsilon(S|T; \alpha) := \mathbb{E}_{S' \sim S \cup T}[R_{emp}(\mathcal{A}(S'); S)] \tag{1}$$

*where the expectation is over both the randomness in $\mathcal{A}$ and $S'$, a random variable that samples a uniformly random $\alpha$ fraction per class (rounded up) from $S \cup T$. When $T = \emptyset$ we also write $\epsilon(S; \alpha)$.*

Here $\alpha$ is a hyperparameter that is typically selected *a priori* on the basis of some domain knowledge (for example, if $\alpha = 1$ and $\mathcal{A}$ trains a deep neural network, then the network can memorize the training set). A typical value of $\alpha$ is $1/4$ or $1/2$; in our ablation studies, we find that our particular application is robust to a wide range of $\alpha$.

The self-expansion property measures the ability of the learning algorithm $\mathcal{A}$ to generalize from a subset of $S \cup T$ to the empirical distribution of $S$. Intuitively, when $T = \emptyset$, a smaller expansion error means that the set $S$ is both "easier" and "more homogeneous" with respect to the learning algorithm, as a random subset is sufficient to learn the contents of $S$. When $T$ is not empty, we expect that the self-expansion error will be lower when $T$ contains similar data as $S$ (for instance, if both $S$ and $T$ are drawn from the same distribution $\mathcal{D}$).

This observation leads directly to our formal definition of compatibility between sets:

**Definition 2.2** (Compatibility of sets.). *A set $T$ is $\alpha$-compatible with set $S$ if*

$$\epsilon(S|T; \alpha) \leq \epsilon(S; \alpha). \tag{2}$$

*Conversely, $T$ is $\alpha$-incompatible with $S$ if the opposite holds, i.e.,*

$$\epsilon(S; \alpha) \leq \epsilon(S|T; \alpha). \tag{3}$$

*Furthermore, $T$ is* completely $\alpha$-compatible *with $S$ if every subset of $T$ is $\alpha$-compatible with every subset of $S$, and similarly for complete incompatibility.*

In other words, $T$ is (formally) compatible with $S$ if $T$ improves the ability of $\mathcal{A}$ to generalize to $S$.

## 2.3 SEPARATION OF INCOMPATIBLE DATA

Our next result allows us to separate subpopulations of data satisfying the incompatibility property (proof in Appendix A). The main idea is that, given any set, we can always achieve better self-expansion by removing incompatible data (if it exists).

**Theorem 2.3** (Sets minimizing expansion error are homogeneous.). *Let $S = A \cup B$ be a set with $A$ and $B$ nonempty and mutually completely $\alpha$-incompatible. Define*

$$\epsilon^* := \min_{S' \subseteq S} \epsilon(S'; \alpha), \tag{4}$$

*let $\mathcal{S}^*$ be the collection of subsets of $S$ that achieve $\epsilon^*$. Then for any* smallest *subset $S^*_{min} \in \mathcal{S}^*$, either $S^*_{min} \subseteq A$ or $S^*_{min} \subseteq B$. Furthermore, if at least one of the incompatibilities is strict, then for* all *$S^* \in \mathcal{S}^*$ we have either $S^* \subseteq A$ or $S^* \subseteq B$.*

Theorem 2.3 suggests an iterative procedure for separating subsets of data that satisfy the incompatibility property. Namely, at each step $i$, identify $S^*_{min}$ in the current dataset $D_i$, then repeat the procedure with $D_{i+1} = D_i \setminus S^*_{min}$. The procedure terminates when $D_{i+1} = \emptyset$. Assuming the subpopulations in the dataset satisfy the incompatibility property, this process partitions the training set into subsets that are guaranteed to contain data from a single subpopulation. Furthermore, if subpopulations satisfy strict incompatibility, we can instead take the largest $S^* \in \mathcal{S}^*$ at each step. The next section develops this insight into a practical clustering mechanism for incompatible subpopulations.

---

**Algorithm 1** Inverse Self-Paced Learning

---

**Input:** training set $S$, total iterations $N$, annealing schedule $1 \geq \beta_0 \geq ... \geq \beta_N = \beta_{\min} > 0$, expansion $\alpha \leq 1$, momentum $\eta \in [0, 1]$, learning algorithm $\mathcal{A}$, initial parameters $\theta_0$
**Output:** $S_N \subseteq S$ such that $|S_N| = \beta_{\min}|S|$
  1: $S_0 \leftarrow S$
  2: $L \leftarrow \mathbf{0}$
  3: **for** $t = 1$ **to** $N$ **do**
  4:     $S' \leftarrow \text{SAMPLE}(S_{t-1}, \alpha)$
  5:     $\theta_t \leftarrow \mathcal{A}(S', \theta_{t-1})$
  6:     $L \leftarrow \eta L + (1 - \eta)R_{emp}(\theta_t; S)$
  7:     $S_t \leftarrow \text{TRIM}(L, \beta_t)$
  8: **end for**

---

## 2.4 INVERSE SELF-PACED LEARNING

A major question raised by Theorem 2.3 is how to identify the set $S^*$. In general, even computing a single self-expansion error of a given set $S$ is intractable as it involves taking the expectation over all subsets of size $\alpha|S|$. The *Inverse Self-Paced Learning* (**ISPL**) algorithm presented below solves this intractibilty problem. Rather than optimizing over all possible subsets of the training data, we instead seek to minimize the expansion error over subsets of fixed size $\beta|S|$. The new optimization objective is defined as:

$$S_\beta^* := \arg \min_{S' \subseteq S : |S'| = \beta|S|} \epsilon(S'; \alpha) \tag{5}$$

We alternate between optimizing parameters $\theta_t$ and the training subset $S_t$—given $S_{t-1}$, we update $\theta_t$ using a single subset from $S_{t-1}$ of size $\alpha|S_{t-1}|$. Then we use $\theta_t$ to compute the loss for each element in $S$, and set $S_t$ to be the $\beta$ fraction of the samples with the lowest losses. To encourage stability of the learning algorithm, the losses are smoothed by a momentum term $\eta$. To encourage more global exploration in the initial stages, we anneal the parameter $\beta$ from an initial value $\beta_0$ down to the target value $\beta_{\min}$. Algorithm 1 presents the full algorithm. SAMPLE takes a set $S$ and samples an $\alpha$ fraction of each class uniformly at random; TRIM takes losses $L$ and returns the $\beta$ fraction with the lowest loss.

To conclude, we show for certain choices of parameters that Algorithm 1 converges to a local optimum of the following objective over the training set $S$:

$$F(\theta_t, S_t; \beta_t) := \sum_{i \in S_t} L(f_{\theta_t}(x_i), y_i) + \max(0, \beta_t|S| - |S_t|) \tag{6}$$

where $S_t$ is a subset of $S$ and $\beta_t$ is decreasing in $t$ (proof in Appendix A).

**Proposition 2.4.** *Set $\alpha = 1$ and $\eta = 0$ in Algorithm 1, and assume that $\mathcal{A}$ returns the empirical risk minimizer. Then we have that for each round of the algorithm, $F(\theta_t, S_t; \beta_t)$ is decreasing in $t$ and furthermore, $|F(\theta_t, S_t; \beta_t) - F(\theta_{t+1}, S_{t+1}; \beta_{t+1})| \xrightarrow{t \to \infty} 0$.*

## 3 DEFENDING AGAINST BACKDOOR ATTACKS

We present a novel characterization of poisoning attacks based on the formal definition of compatibility introduced in Section 2. In our threat model, the attacker aims to affect the trained model's behavior in ways that do not emerge from clean data alone, so we characterize the poisoned data by its incompatibility with clean data. We prove this observation holds for a backdoor attack in the context of linear classifiers. As Theorem 2.3 yields a method of partitioning the dataset into clusters of clean and poisoned data (when the clean and poisoned data are incompatible), we present a boosting algorithm to *identify* which clusters contain clean data (and hence remove the poisoned data).

### 3.1 THREAT MODEL

The attacker's goal is to install a backdoor in the trained network that 1) changes the poisoned network's predictions on poisoned data while 2) preserving the accuracy of the poisoned network on

clean data. For the backdoor to be meaningful, the behavior of a poisoned model on poisoned data should also be inconsistent with a model trained on clean data.

To begin, a set of $n$ clean training samples $D$ is drawn iid from $\mathcal{D}$. After observing $D$, the attacker selects at most $n/2 - 1$ samples $A \subseteq D$ (leaving behind at least $n/2$ clean samples $C = D \setminus A$) and fixes a pair of perturbation functions $\tau_{train} : X \times Y \mapsto X \times Y$ (the attacker uses $\tau_{train}$ to construct the poisoned dataset for training, changing the image and potentially the label of each poisoned element) and $\tau_{test} : X \mapsto X$ (the attacker uses $\tau_{test}$ to construct poisoned test images without changing the ground-truth label of the poisoned image). The attacker then poisons the samples in $A$ to obtain poisoned samples $P = \{\tau_{train}(a) \mid a \in A\}$. The threat model places no restrictions on $\tau_{train}$, but requires that $\tau_{test}$ should not affect the predictions of a network trained on $C$. The final poisoned training set is $C \cup P$.

Let $\theta$ be the parameters of the network under attack. Given $m$ fresh test samples $T$ drawn iid from $\mathcal{D}$, the attacker seeks to maximize the *targeted misclassification rate* (TMR):

$$\text{TMR}(\theta; T) := \frac{1}{m} \sum_{i=1}^{m} L(f_\theta(x_i), \neg y_i) * L(f_\theta(\tau_{test}(x_i)), y_i), \tag{7}$$

i.e., the attack succeeds if it can flip the label of a correctly-classified instance by applying the trigger $\tau_{test}(\cdot)$ at test time. This metric captures both the "hidden" nature of the backdoor (the first term rewards the attacker only if clean inputs are correctly classified) as well as control upon application of the trigger (the second term rewards the attacker only if poisoned inputs are misclassified).

### 3.1.1 DEFENSE CAPABILITIES AND OBJECTIVE

The defender is given only the poisoned dataset $C \cup P$. In particular, $\tau_{train}$, $\tau_{test}$, $A$, and $C$ are not available to the defender. The defender's objective is to return a sanitized dataset $\tilde{D} \subseteq C \cup P$ such that the learned parameters $\tilde{\theta} := \mathcal{A}(\tilde{D})$ are as close as possible to the ground-truth parameters $\theta^* := \mathcal{A}(C)$. Note that $\theta^*$ is guaranteed to exhibit low targeted misclassification rate—the threat model requires that for fresh $(x, y)$ drawn from $\mathcal{D}$, $f_{\theta^*}(\tau_{test}(x)) = y$ with high probability.

### 3.2 CHARACTERIZING BACKDOOR POISONING ATTACKS USING INCOMPATIBILITY

The following property formalizes the intuition that increasing the amount of clean data in the training set should not improve the performance on poisoned data (and vice versa):

**Definition 3.1** (Incompatibility Property). *Let $C$ and $P$ be the clean and poisoned data produced by an attacker according to threat model defined in Section 3.1. We say that the attacker satisfies the* incompatibility property *if $C$ and $P$ are mutually completely incompatible. Furthermore if at least one of the incompatibilities between $C$ and $P$ is strict, then the attacker satisfies the* strict incompatibility property.

Note that incompatibility is defined with respect to the learning algorithm $\mathcal{A}$ as well as the sub-sampling rate $\alpha$. To motivate this definition, we exhibit a backdoor attack against linear classifiers which falls under the threat model in Section 3.1 and provably satisfies the incompatibility property:

**Theorem 3.2.** *Let the input space be $X = \mathbb{R}^N$ ($N > 1$). We use the parametric family of functions $f_{w,b}$ consisting of linear separators given by $f_{w,b}(x) = \text{sign}(\langle w, x \rangle + b)$ for $w \in \mathbb{R}^N, b \in \mathbb{R}$. $\mathcal{A}$ selects the maximum margin classifier. Then there exists a distribution $\mathcal{D}$ and perturbation functions $\tau_{train}$ and $\tau_{test}$, such that if the adversary poisons $1/4$ of each class at random, then with high probability over the initial training samples:*

*1. Given the clean dataset, $\mathcal{A}$ returns parameters with 0 empirical (and population) risk.*

*2. Given the poisoned dataset, $\mathcal{A}$ returns parameters with a targeted misclassification rate of 1.*

*3. The clean and poisoned data are mutually completely incompatible.*

The proof is in Appendix A. The idea is to construct a dataset such that there exist dimensions in the original, clean feature space that are uncorrelated with the true labels, i.e., consist of pure noise. The attacker makes use of these extra dimensions by replacing the noise with a backdoor signal. Although this result uses linear classifiers, the setting is analogous to many existing attacks against DNNs; for example, many backdoor attacks against image classifiers insert synthetic patches on the border of the image, which is a location that does not typically affect the original classification task.

---

**Algorithm 2** Boosting Homogeneous Sets

---

**Input:** Homogeneous sets $S_1, ..., S_N$, number of samples $n = \sum_{i=1}^{N} |S_i|$, learning algorithm $\mathcal{A}$
**Output:** Votes $V_1, ..., V_N$

 1: $C_1, ..., C_N \leftarrow 0$
 2: **for** $i = 1$ **to** $N$ **do**
 3:    $\theta_i \leftarrow \mathcal{A}(S_i)$
 4:    **for** $k = 1$ **to** $N$ **do**
 5:      **if** $\text{Loss}_{0,1}(\theta_i; S_k) < |S_k|/2$ **then**
 6:        $C_k \leftarrow C_k + |S_i|$
 7:      **end if**
 8:    **end for**
 9: **end for**
10: **for** $i = 1$ **to** $N$ **do**
11:    $V_i \leftarrow C_i > n/2$
12: **end for**

---

### 3.3 Identification using Boosting

We conclude this section by showing how to identify the clean data given the output of our clustering mechanism, which is a collection of homogeneous (i.e., entirely clean or entirely poisoned) subsets. One idea is to measure the compatibility between each pair of subsets, then take the largest mutually compatible (sub)collection of subsets (under the assumption that the clean data is always compatible with itself). However in general this requires training on each subset multiple times in sequence. We instead propose a simplified boosting algorithm for identifying the clean data which involves training on each subset exactly once. The main idea is to fit a weak learner to each cluster, then use each weak learner to vote on the other clusters.

The following sufficient conditions guarantee the success of our boosting algorithm:

**Property 3.3** (Compatibility property of disjoint sets). *Let $C'$ and $C''$ be any two disjoint sets of clean data, and let $P'$ be a set of poisoned data. Then*

$$R_{emp}(\mathcal{A}(C'); C'') < 1/2 \leq R_{emp}(\mathcal{A}(C'); P'). \tag{8}$$

The first inequality says that training and testing on disjoint sets of clean data is an improvement over random guessing. The second inequality says that training with clean data then testing on poisoned data is at best equivalent to random guessing; this property is natural in that the threat model already guarantees $R_{emp}(\mathcal{A}(C); P)$ is large (i.e., close to 1) for *random* poisoned data $P$.

Algorithm 2 presents our approach for boosting from homogeneous sets. The subroutine $\text{Loss}_{0,1}$ takes a set of parameters and a set $S$, and returns the total risk over $S$ using the zero-one loss. Each subset's vote is weighted by its size. We have the following correctness guarantee (proof in Appendix A):

**Theorem 3.4** (Identification of clean samples). *Let $S_1, ..., S_N$ be a collection of homogeneous subsets containing either entirely clean or entirely poisoned data. Then if the clean and poisoned data satisfy Property 3.3, Algorithm 2 votes $V_i = \texttt{True}$ if $S_i$ is clean (and $V_i = \texttt{False}$ otherwise).*

## 4 Experimental evaluation

We report empirical evaluations of the incompatibility property (Section 4.1) and the proposed defense (Section 4.2). We use the CIFAR-10 dataset (Krizhevsky & Hinton, 2009) for general image recognition, containing 10 classes of 5000 training and 1000 test images each, and the GTSRB dataset (Stallkamp et al., 2012) for traffic sign recognition, containing 43 classes with 26640 training and 12630 test images, respectively. The GTSRB training set is highly imbalanced, with classes ranging from 150 to 1500 instances.

Our experiments consider three types of backdoor attacks. Dirty label backdoor (DLBD) attacks use small patches ranging from a single pixel to a 3x3 checkerboard pattern, following the implementation of Tran et al. (2018). WATERMARK uses 8x8 images such as a copyright sign or a peace

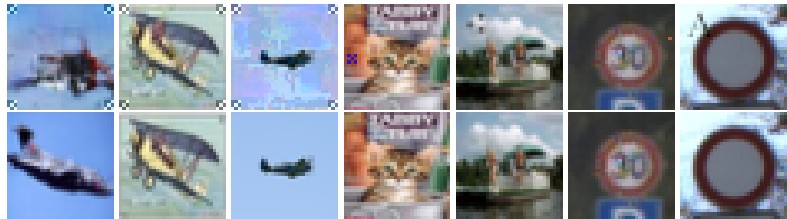

Figure 1: Pairs of poisoned (top) and clean (bottom) samples for all datasets, selected for visible triggers. From left to right: CIFAR-10 with CLBD using GAN, $\ell_2$, and $\ell_\infty$ perturbations, DLBD, and WATERMARK attacks; and GTSRB with DLBD and WATERMARK attacks.

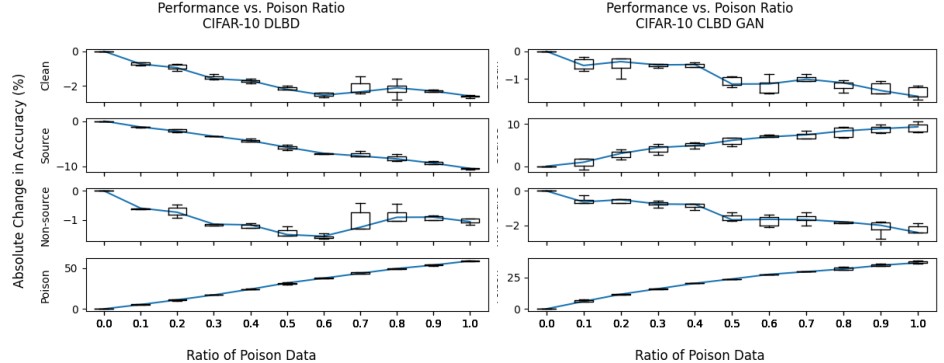

Figure 2: Results from incompatibility experiments on CIFAR-10, DLBD (left) and CLBD (right). Each subplot shows how accuracy changes over different subsets of data as the amount of poison increases. The top plot displays change in accuracy on clean training data, i.e., $\epsilon(C|P; \alpha) - \epsilon(C; \alpha)$; the next two plots separate the change in accuracy on $C$ into source and non-source classes; the bottom plot shows change in accuracy on poisoned data $P$. All axes use linear scales.

sign, taken from Gao et al. (2019). Both of these attacks insert a patch into a training image from a source class, then change the training label to the target class. The attacker's objective is to induce the learner to misclassify images of the source class as the target class upon application of the patch. Clean label backdoor (CLBD) attacks use the poisoned datasets from Turner et al. (2018), which are only available for CIFAR-10. Each attack first strongly perturbs each image of the source class independently using either GAN, $\ell_2$, or $\ell_\infty$ bounds, then inserts several patches (without changing the label); the attacker's objective is to make the learner misclassify an image from *any non-source class* as the source class by applying the patches. Figure 1 presents examples from each dataset. Additional dataset construction details can be found in Appendix B. All experiments presented in the main text use a PreActResNet-18 architecture. Full experimental details and additional results, including performance against an adaptive white-box attacker and experiments testing the sensitivity of ISPL to its hyperparameters, are contained in Appendices C and D.

## 4.1 INCOMPATIBILITY BETWEEN CLEAN AND POISONED DATA

We evaluate whether the attacks satisfy the incompatibility property by measuring the gap between $\epsilon(C; \alpha)$ and $\epsilon(C|P; \alpha)$ for clean data $C$ and poisoned data $P$ generated by the 3 attacks (DLBD, WATERMARK, CLBD) on CIFAR-10. Figure 2 presents selected plots for the DLBD (e.g., relabelling airplanes as birds) and GAN-based CLBD (e.g., perturbing airplanes toward other classes) scenarios. Along the x-axis, we increase the ratio of poisoned to clean data in the source class from 0 to 1. Our experimental procedure is as follows: we first sample $1/8$ of each class in the clean dataset to create $C$. For each poison ratio, we sample $P' \sim P$ according to the poison ratio, train on $P' \cup C$ with $\alpha = 1/2$, then take the mean over 5 samples of $P'$ as the self-expansion error. The entire process is repeated for 5 samples of $C$ for each poisoned dataset in our evaluations (8 total for DLBD, 5 total for GAN-based CLBD). To better present the gap $\epsilon(C|P; \alpha) - \epsilon(C; \alpha)$, we report

Table 1: Summary of performance for **ISPL+B** across several datasets and poisoning strategies. Columns display different datasets and poisoning strategies (e.g., poisoning CIFAR-10 using the CLBD attack). The rows display different amounts of poison (e.g., $\epsilon = 5$ refers to $5\%$ of the source class being poisoned). An entry of the form "7 / 8" indicates that **ISPL+B** successfully defended 7 of the 8 scenarios in the corresponding setting. Please refer to Section 4.2 for a detailed description.

| | | CIFAR-10 | | | | GTSRB | | totals |
|---|---|---|---|---|---|---|---|---|
| | | DLBD | | WATERMARK | CLBD | DLBD | WATERMARK | |
| | 1-to-1 | all-to-1 | all-to-all | | | | | |
| $\epsilon$   5 | 7 / 8 | 4 / 4 | 4 / 4 | 6 / 8 | 15 / 15 | 6 / 8 | 8 / 8 | 50 / 55 |
| 10 | 8 / 8 | 4 / 4 | 2 / 4 | 5 / 8 | 15 / 15 | 6 / 8 | 8 / 8 | 48 / 55 |
| 20 | 7 / 8 | 4 / 4 | 2 / 4 | 1 / 8 | 11 / 15 | 5 / 8 | 6 / 8 | 36 / 55 |
| totals | 22 / 24 | 12 / 12 | 8 / 12 | 12 / 24 | 41 / 45 | 17 / 24 | 22 / 24 | 134 / 165 |

each metric as an absolute difference in accuracy from the unpoisoned case (ratio = 0). A trendline connects the medians.

Our first observation is that the key quantity, $\epsilon(C|P; \alpha) - \epsilon(C; \alpha)$, is negative for all poisoned datasets, which empirically supports our insight that clean and poisoned data are incompatible (or even strictly incompatible), despite the qualitative difference behind the attack mechanisms. The trend in the top plot also indicates that the gap between $\epsilon(C|P; \alpha)$ and $\epsilon(C; \alpha)$ increases with the amount of poisoned data. The next two plots show how the gap breaks down over the source and non-source classes. For the DLBD attack, the source class shows a strong negative trend in accuracy as the amount of poison increases, while accuracy on the non-source classes stays largely constant. This fact is consistent with the hypothesis that the poisoned data is impairing the accuracy of the classifier on clean data from the source class (e.g., clean airplanes are getting misclassified as birds).

For the CLBD attack, even though the total clean accuracy decreases, the source class accuracy actually increases as the amount of poison increases. We attribute this phenomenon to the classifier learning the perturbed poisoned training instances in the source class, which yields improved accuracy on the clean instances as well. The decrease in clean accuracy is therefore due to a drop in accuracy among the non-source classes. Because the CLBD attack produces perturbations of the source class toward other classes (thus making the other classes more similar to, e.g., airplanes), this behavior is consistent with the classifier misclassifying clean instances of non-source classes as the source class. These observations are corroborated by the increase in accuracy on poisoned data in both cases, suggesting that the classifier's mistakes on clean data are correlated with increased accuracy on similar-looking poisoned data. Appendix D contains further discussion and results.

## 4.2 PERFORMANCE OF PROPOSED DEFENSE

We next evaluate the performance of our proposed defense on the full range of attacks. Most settings use one-to-one attacks (e.g., place a trigger on dogs and mislabel as cats; an $\epsilon$ percentage of the source class is poisoned), which is the most common setting in the literature (Gu et al., 2019). For the CIFAR-10 DLBD scenario, we additionally use all-to-one (e.g., place a trigger on all non-cats and mislabel as cats; an $\epsilon/9$ percentage of each non-target class is poisoned) and all-to-all attacks (i.e., place a trigger on any image and mislabel according to a cyclic permutation on classes; an $\epsilon$ percentage of every class is poisoned).

Table 1 presents a summary of our results. Each entry is of the format "success / total", where total indicates the total number of different scenarios for the attack setting (i.e., different source and target class and trigger combinations), and a run is successful when the defense achieves TMR below 1%. We note that the clean accuracy is around 91% on CIFAR-10 and 94% on GTSRB after running **ISPL+B**, compared to around 93% and 94% when training on the full, unpoisoned datasets, respectively. These results indicate that our approach succeeds in defending against various backdoor poisoning attacks, particularly for the standard settings of $\epsilon = 5, 10$ in the literature (Tran et al., 2018), for a small cost in clean accuracy. Appendix D contains the unabridged results.

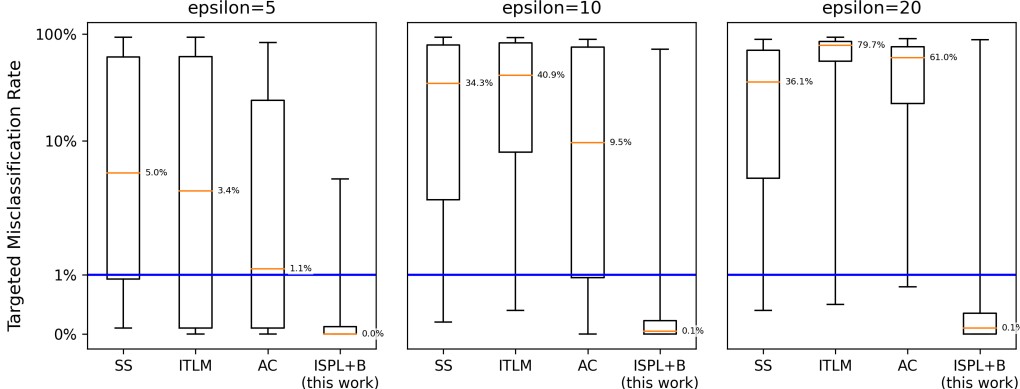

Figure 3: Comparison with baseline defenses on the CIFAR-10 DLBD (1-to-1) scenario for $\epsilon = 5, 10, 20$. Each plot displays TMRs (lower is better) over 24 runs. The orange lines indicate the median TMR, the boxes indicate the top and bottom quartile, and the whiskers cover the entire range of performance. The blue line displays the cutoff for success used in Table 1 (i.e., TMR <1%).

**Comparison with Existing Approaches.** Figure 3 compares the performance of our defense with 3 existing defenses—Spectral Signatures (Tran et al., 2018), Iterative Trimmed Loss Minimization (Shen & Sanghavi, 2019), and Activation Clustering (Chen et al., 2018)—on the CIFAR-10 DLBD (1-to-1) scenario. **ISPL+B** outperforms all 3 baselines in terms of reducing the TMR. Table 2 in Appendix D contains the detailed results of this experiment.

## 5 RELATED WORK

Many prior works propose methods for defending against backdoor attacks on neural networks. One common strategy, which can be viewed as a type of deep clustering (Zhou et al., 2022), is to first fit a deep network to the poisoned distribution, then partition on the learned representations. The Activation Clustering defense (Chen et al., 2018) runs k-means clustering (k=2) on the activation patterns, then discards the smallest cluster. Tran et al. (2018) propose a defense based on the assumption that clean and poisoned data are separated by their top eigenvalue in the spectrum of the activations. TRIM (Jagielski et al., 2021) and ITLM (Shen & Sanghavi, 2019) iteratively train on a subset of the data created by removing samples with high loss. However, each iteration removes the same small number of samples; both direct comparisons with ITLM and our ablation studies suggest that the dynamic resizing in ISPL is crucial to identifying the poison. Shan et al. (2022) is motivated by a similar intuition as our compatibility property; however their formalization differs significantly from ours, and they do not present a defense but rather a forensics technique, which is only able to identify the remaining poison given access to a known poisoned input. Finally, several works provide certified guarantees against data poisoning attacks (Steinhardt et al., 2017; Jia et al., 2022); in exchange for the stronger guarantee, these works defend a small fraction of the dataset, e.g., the state-of-the-art certifies 16% of CIFAR-10 against 50 poisoned images (Wang et al., 2022a). We refer the reader to Wang et al. (2022b) for a comprehensive survey of data poisoning defenses. Appendix E contains a discussion of other related works.

## 6 CONCLUSION

Backdoor data poisoning attacks on deep neural networks are an emerging class of threats in the growing landscape of deployed machine learning applications. We present a new, state-of-the-art defense against backdoor attacks by leveraging a novel clustering mechanism based on an incompatibility property of the clean and poisoned data. Because our technique works with any class of trained models, we anticipate that our tools may be applied to a wide range of datasets and training objectives beyond data poisoning. More broadly, this work develops an underexplored perspective based on the interaction between the data and training process.

ACKNOWLEDGMENTS

We gratefully acknowledge support from DARPA Grant HR001120C0015. The views expressed are those of the authors and do not reflect the official policy or position of the Department of Defense or the U.S. Government.

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

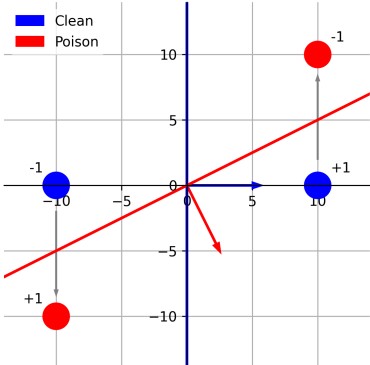

Figure 4: The construction used in our proof of Theorem 3.2, for $N = 2$ dimensions. The balls depict the support of the clean training distribution (blue) and the poisoned data (red). The perturbation function $\tau$ is shown by the grey arrows. The lines are the ground truth maximum margin classifier (blue) and the maximum margin classifier when training with a mixture of clean and poisoned data (red). The red and blue arrows point in the direction of half-plane labelled by the corresponding classifier as the positive class.

## A    DEFERRED PROOFS

*Proof of Theorem 3.2.* First we give a construction when the dimension $N = 2$. We define the training distribution as $\mathcal{D} = (\mathcal{D}_+ + \mathcal{D}_-)/2$, where $\mathcal{D}_+$ and $\mathcal{D}_-$ are both distributed arbitrarily within a ball of radius 1 (e.g., as a truncated Gaussian); $\mathcal{D}_+$ is centered at $(10, 0)$ with labels $+1$ and $\mathcal{D}_-$ is centered at $(-10, 0)$ with labels $-1$. The adversary chooses the perturbation function $\tau$ which flips the label during training and sets the last entry to be $10$ if the original label was $+1$, and $-10$ if the original label was $-1$. Note that $\tau$ is bounded in the sense that as $N \to \infty$, the (expected) magnitude of a data point can grow as $O(\sqrt{N})$ while the magnitude of the perturbation is $O(1)$. Figure 4 depicts the construction.

Clearly, one sample from each clean class is sufficient to learn a maximum margin classifier (MMC) achieving zero population risk, i.e., the MMC will be close to $w = (1, 0)$, $b = 0$.

Next, we show that the attacker succeeds under the threat model in Section 3.1. Recall that the attacker selects 1/4 of the positive samples and 1/4 of the negative samples to poison. Given a poisoned dataset containing at least one example of each class (both clean and poisoned), the classifier given by $w = (1, -2)$, $b = 0$ achieves perfect accuracy on both the clean data and the poisoned data, and hence the MMC also has a poison misclassification rate of 1.

We now show that this setup satisfies the incompatibility property defined in Section 3.2. We begin by showing that the poisoned data is not compatible with the clean data. Let $C$ be any set of clean data, and let $P$ be any set of poisoned data. We have two cases. First, if $C$ contains samples of both classes, then by definition, any subset $C'$ of $C$ in the measurement of the self-expansion error will also include at least one sample of each clean class. Hence, the MMC learned from $C'$ will be close to $w = (1, 0)$, $b = 0$, which achieves perfect accuracy on all clean data. On the other hand, if $C$ contains only samples of the positive class, the MMC is $w = (0, 0)$, $b = +1$, and again all samples in $C$ are correctly classified. The same argument works when $C$ contains only samples from the negative class. Thus, in neither case can poisoned data improve the self-expansion error of $C$.

As the exact same line of reasoning holds for proving incompatibility of clean data with respect to poisoned data, we omit the proof. Therefore the clean and poisoned data are mutually completely incompatible.

To generalize this to $N > 2$, we can simply embed $\mathcal{D}$ in the first two dimensions of the space. The entire construction can also be scaled then translated and rotated without affecting any of the conclusions. □

Before moving on to the proof of Theorem 2.3, we first prove a useful lemma:

**Lemma A.1.** *Let $A$ and $B$ be arbitrary sets, and define $S = A \cup B$. Then for all $\alpha$,*

$$|S|\epsilon(S;\alpha) = |A|\epsilon(A|B;\alpha) + |B|\epsilon(B|A;\alpha) \tag{9}$$

*with the convention that $\epsilon(\emptyset|T;\alpha) = 0$ for all $T$ and $\alpha$.*

*Proof.* Notice that all the expansion errors sample from the same training set $S = A \cup B$. The lemma then follows from the linearity of the unnormalized empirical risk in the test set, i.e.,

$$|S|\epsilon(S;\alpha) = |S|\mathbb{E}_{S'\sim S}\Big[R_{emp}(\mathcal{A}(S');S)\Big] \tag{10}$$

$$= |S|\mathbb{E}_{S'\sim S}\Big[|S|^{-1}\sum_{(x_i,y_i)\in S} L(f_{\mathcal{A}(S')}(x_i),y_i)\Big] \tag{11}$$

$$= \mathbb{E}_{S'\sim S}\Big[\sum_{(x_i,y_i)\in A} L(f_{\mathcal{A}(S')}(x_i),y_i) + \sum_{(x_i,y_i)\in B} L(f_{\mathcal{A}(S')}(x_i),y_i)\Big] \tag{12}$$

$$= \mathbb{E}_{S'\sim S}\Big[|A|R_{emp}(\mathcal{A}(S');A)\Big] + \mathbb{E}_{S'\sim S}\Big[|B|R_{emp}(\mathcal{A}(S');B)\Big] \tag{13}$$

$$= |A|\epsilon(A|B;\alpha) + |B|\epsilon(B|A;\alpha), \tag{14}$$

as claimed. □

*Proof of Theorem 2.3.* Assume first that the incompatibility is not strict. Define $U = S^*_{min} \cap A$ and $V = S^*_{min} \cap B$. From Theorem A.1, we have that

$$|S^*_{min}|\epsilon^* = |U|\epsilon(U|V;\alpha) + |V|\epsilon(V|U;\alpha). \tag{15}$$

Applying now the definition of incompatibility gives

$$|S^*_{min}|\epsilon^* \geq |U|\epsilon(U;\alpha) + |V|\epsilon(V;\alpha). \tag{16}$$

As $|U| + |V| = |S^*_{min}|$, it follows that at least one of $\epsilon(U;\alpha)$ or $\epsilon(V;\alpha)$ is at most $\epsilon^*$, which contradicts the optimality of $S^*_{min}$ (as it was assumed to be a smallest set achieving $\epsilon^*$).

Now we consider the case when at least one of the incompatibilities is strict. Let $S^*$ be any set in $\mathcal{S}^*$. Using the definition of strict incompatibility, we get that

$$|S^*|\epsilon^* > |U|\epsilon(U;\alpha) + |V|\epsilon(V;\alpha). \tag{17}$$

Hence, at least one of $\epsilon(U;\alpha)$ or $\epsilon(V;\alpha)$ is strictly less than $\epsilon^*$, which again contradicts the optimality of $S^*$. □

*Proof of Theorem 3.4.* Let $S_i$ and $S_j$ be a clean component and poison component, respectively. By the second inequality in Property 3.3, we have that $R_{emp}(\mathcal{A}(S_i);S_j) \geq 1/2$. Thus $S_i$ votes `False` on $S_j$. Conversely, if both $S_i$ and $S_j$ are a clean components, then $R_{emp}(\mathcal{A}(S_i);S_j) < 1/2$, so $S_i$ votes `True` on $S_j$. Putting these together and using the fact that at least half of the data is clean, we find that the poisoned components have weighted vote strictly less than $|S|/2$, while the clean components have weighted vote strictly greater than $|S|/2$. As all the clean components vote correctly, using the weighted majority correctly identifies all the clean and poisoned components as claimed. □

*Proof of Proposition 2.4.* Recall that $\alpha = 1$ and $\eta = 0$. We prove the statement in two steps.

First, we show that

$$F(\theta_{t+1}, S_t; \beta_t) \leq F(\theta_t, S_t; \beta_t) \tag{18}$$

The inequality follows from the optimization on Line 5 in Algorithm 1, which sets $\theta_{t+1}$ to the empirical risk minimizer of the set $S_t$.

Next, we claim that

$$F(\theta_{t+1}, S_{t+1}; \beta_{t+1}) \leq F(\theta_{t+1}, S_t; \beta_t) \tag{19}$$

Since $|L(\cdot, \cdot)| \leq 1$, the optimal size of the set $S_t$ is $|S_t| = \beta_t |S|$. Since $\beta_t$ is decreasing, we have that $|S_{t+1}| \leq |S_t|$. Thus the number of elements in the trimmed empirical loss is non-increasing (Line 7, Algorithm 1).

Combining the two inequalities shows that the objective function is decreasing in $t$. Since $F(\theta_t, S_t; \beta_t)$ is a decreasing sequence bounded from below by zero, the monotone convergence theorem gives the second result.

$\square$

## B  DATASET CONSTRUCTION DETAILS

Our implementation of the 1-to-1 dirty label backdoor (DLBD) adversary follows the threat model described in Gu et al. (2017). For evaluation, we use the same dataset (CIFAR-10 (Krizhevsky & Hinton, 2009)) and setup for our experiments as the Spectral Signatures work (Tran et al., 2018). Each scenario has a single source and target class, and we use the same (source, target) pairs as in Tran et al. (2018): (airplane, bird), (automobile, cat), (bird, dog), (cat, dog), (cat, horse), (horse, deer), (ship, frog), (truck, bird).

The perturbation function $\tau$ overlays a small pattern on the image at a fixed location. All patterns fit within a 3x3 pixel box. To generate a perturbation, we choose a shape (L-shape, X-shape, or pixel) uniformly at random. The (x,y) coordinates of the perturbation are randomly selected to guarantee that the entire shape is visible before data augmentation (e.g., the pixel-based perturbation can be placed anywhere within the 32x32 image, but the X-shape is larger and so must be centered in a 30x30 region, one pixel away from the border). The color of the perturbation is also selected uniformly at random, with each of the (R,G,B) coordinates ranging from 0 to 255. Finally, we randomly select an $\epsilon = 5, 10, 20\%$ percentage of the source class, apply the perturbation by replacing the pixels in the corresponding locations with the selected shape and color, then relabel the poisoned images as the target class. For the all-to-1 case, for a given target class, we poison an equal proportion of every non-target class such that the total amount of poison is $\epsilon$ times the size of the source class. For the all-to-all case, we select a cyclic permutation of the classes, then poison a $\epsilon$ percentage of each class.

The construction of the WATERMARK dataset is the same, except that we instead use 8x8 images depicting: a peace sign, the letter A, 3 bullet holes, or a colorful patch, taken from Nicolae et al. (2019). Each trigger has a transparent background, and is blended into the upper left corner of the original image with 80% opacity (where 0% opacity would return the original image without the trigger, and 100% opacity would superimpose the trigger on top of the original image).

For the CLBD dataset, we used the official datasets provided by Turner et al. (2018). There are three datasets in total, one for each perturbation type: $\ell_2$, $\ell_\infty$, and GAN. For the $\ell_2$ and $\ell_\infty$-based attacks, each image is perturbed independently to maximize the loss with respect to a reference model trained on clean data; the perturbation size is bounded by the respective norm. For the GAN-based attack, for each image to be poisoned, a perceptually similar target image in another class is identified using the latent space of the GAN, and the perturbed image is an interpolation of the original and target images in the latent space. In all three cases, a 3x3 checkerboard patch is then placed in each of the 4 corners of the image. The label of the image is not changed. At test time, only the checkerboard patches are placed on the image. We refer the reader to Turner et al. (2018) for more details.

For the GTSRB dataset, we use the same construction of the DLBD and WATERMARK attacks on CIFAR-10. As the CLBD dataset was only created for the CIFAR-10 dataset, we did not evaluate the CLBD attack on GTSRB.

## C  DEFENSE SETUPS AND HYPERPARAMETERS

**ISPL + Boosting (this work).**    For our defense, we use the set of hyperparameters described here unless otherwise noted. For CIFAR-10, we run 8 rounds of ISPL, each of which returns a component consisting of roughly 12% of the total samples. For GTSRB, we run 4 rounds of ISPL so that each component contains 24% of the total samples. Let $p$ be the target percentage of samples over the remaining samples (e.g., for CIFAR-10 $p \approx 1/(8 - i + 1)$ in the $i^{th}$ iteration). Then the number of iterations $N$ is set to $2 + \min(3, 1/p)$. $\beta$ starts at $3 * p$ in the first iteration, then drops linearly to its final value of $p$ over the next 2 iterations. When trimming the training set, we also additionally include the top $p/8$ samples per class to prevent the network from collapsing to a trivial solution. For the learning procedure $\mathcal{A}$, we use standard SGD, trained for 4 epochs per iteration, with a warm-up in the first iteration of 8 epochs. The expansion factor $\alpha$ is set to $1/4$, and the momentum factor $\eta$ is set to $0.9$.

We run ISPL 3 times to generate 24 weak learners for CIFAR-10, and 12 weak learners for GTSRB. For the boosting framework, each weak learner is trained for 40 epochs on its respective subset, then votes on a per-sample basis. The sample is preserved if the modal vote equals the given label, with ties broken randomly, or the sample is in the lower half of its class by loss.

For CIFAR-10, we also include a final self-training step by training a fresh model for 100 epochs on the recovered samples. The main idea is that a model fit to the full "clean" training data can be used to test the excluded training data, thereby recovering additional consistent data which may have been originally excluded because the weak learners were fit to a small subset of data for fewer epochs. However, it may take several repetitions of training a model from scratch before this self-training process no longer identifies new samples to recover. Therefore, we use a simple self-paced learning algorithm to dynamically adjust the samples during training to limit the self-training to a single iteration. More explicitly, we start with the "clean" samples as returned by the boosting framework. Every 5 epochs, we update the training set to be the samples whose labels agree with the model's current predictions. Due to the relative frequency with which we resample the training set, we smooth the predictions by a momentum factor of $0.8$ so that the training process is less noisy. The samples used for training in the last epoch are returned as the defended dataset. In our experiments, this process decreases the false positive rate (and thus increases the clean accuracy) but does not materially affect the false negative rate (nor the targeted misclassification rate). We did not use self-training for GTSRB as we found the clean accuracy was sufficiently high.

**Spectral Signatures.**    We use the official implementation of the Spectral Signatures (**SS**) defense (Tran et al., 2018) by the authors, available on Github, except that we replace the training procedure with PyTorch (instead of Tensorflow 1.x as in the authors' original implementation). The authors suggest removing 1.5 times the maximum expected amount of poison from each class for the defense. We remove 20% of each class for $\epsilon = 5, 10\%$ (to match the procedure in their paper) and 30% of each class for $\epsilon = 20\%$. In selecting the layer for the activations, for the ResNet32 architecture we use the input to the third block of the third layer (taken from the SS defense authors' public implementation), and for the PreActResNet18 architecture we use the input to the first block of the fourth layer (which was found empirically to remove the most poison on the first set of scenarios). We note that the authors indicate the defense should be fairly successful at any of the later layers of the network.

**Iterative Trimmed Loss Minimization.**    The Iterative Trimmed Loss Minimization (**ITLM**) defense (Shen & Sanghavi, 2019) consists of an iterative procedure. Given a setting $0 < \alpha \leq 1$, one first trains a model for a number of epochs. Then the $\alpha$ fraction of samples with the lowest loss are retained for the next iteration. This process is repeated several times, with a fresh model beginning each iteration. The defended dataset is the $\alpha$ fraction of samples with the lowest loss after the last iteration. For the backdoor data poisoning experiments on CIFAR-10, the authors use 80 epochs for the first round of training, then 40 epochs thereafter; they also set $\alpha = 98\%$ for $\epsilon = 5\%$, and do not test at other values of $\epsilon$. We use the same settings, and scale $\alpha$ linearly with $\epsilon$, i.e., $\alpha = 96\%$ for $\epsilon = 10\%$ and $\alpha = 92\%$ for $\epsilon = 20\%$.

**Activation Clustering.**    The Activation Clustering (**AC**) defense (Chen et al., 2018) has an actively maintained official implementation in the Adversarial Robustness Toolbox (ART) (Nicolae et al.,

2019), an open-source collection of tools for security in machine learning. We use the official implementation with the default parameters values in ART v1.6.2. In selecting the layer for the activations, we used the same layers as for Spectral Signatures.

**Models.** For the ResNet32 (He et al., 2016a) model, we use vanilla SGD with learning rate 0.1, momentum 0.9, and weight decay 1e-4. For the final dataset, we train for 100 epochs (unless otherwise noted) and drop the learning rate by 10 at epochs 75 and 90. Using these parameters, we achieve around 94.5% accuracy on GTSRB when trained and tested with clean data.

For the PreActResNet18 (He et al., 2016b) model, we use vanilla SGD with learning rate 0.02, momentum 0.9, and weight decay 5e-4. For the final dataset, we train for 100 epochs (unless otherwise noted) and drop the learning rate by 10 at epochs 50, 75, and 90. Using these parameters, we achieve around 93.0% accuracy on CIFAR-10 when trained and tested with clean data.

For the ResNet56 (He et al., 2016b) model, we use vanilla SGD with learning rate 0.05, momentum 0.9, and weight decay 5e-4. For the final dataset, we train for 100 epochs (unless otherwise noted) and drop the learning rate by 10 at epochs 75, 90, 95, and 98. Using these parameters, we achieve around 91.0% accuracy on Imagenette when trained and tested with clean data.

# D ADDITIONAL EXPERIMENTAL RESULTS

This section contains additional experimental results to complement the main text. Appendix D.1 expands upon the empirical evaluations of the compatibility property from Section 4.1. Appendix D.2 reports comprehensive results from the evaluations of the **ISPL+B** defense reported in Section 4.2. Appendix D.2 also includes a detailed breakdown of the performance for the three defenses (Spectral Signatures, Activation Clustering, and Iterative Trimmed Loss Minimization) we use as baselines in our comparisons. Appendix D.2.1 reports results for our defense against the Sleeper Agent attack (Souri et al., 2022), and Appendix D.2.2 reports results for the DLBD attack on a high-resolution dataset. Appendix D.2.3 evaluates our defense against an adaptive attack that is given white-box access to the ISPL subroutine, and selects which elements to poison based on the behavior of ISPL on clean data. Our results show that **ISPL+B** maintains excellent performance against the adaptive attacker, defending against 20 out of 24 scenarios. Appendix D.2.4 reports results for the CIFAR-10 DLBD attack after training for 200 epochs on both the PreActResNet18 and ResNet32 architectures, to test the robustness of our results to the training setup. Finally, to evaluate the sensitivity of ISPL to the main hyperparameters $\alpha$ and $\beta$, Appendix D.2.5 compares the performance of ISPL on a selection of attack scenarios across a range of $\alpha$ and $\beta$. Our main finding is that performance remains high for a range of choices of $\alpha$ and $\beta$.

All results reported use the median over 3 random seeds, unless otherwise noted.

## D.1 INCOMPATIBILITY EVALUATION

Figures 5-9 displays aggregated results of the empirical study of incompatibility for the CLBD, DLBD, and WATERMARK attacks on CIFAR-10. Plots on the left side measure the compatibility of poisoned with clean data (i.e., $\epsilon(C; \alpha) - \epsilon(C|P; \alpha)$) and plots on the right side measure the compatibility of clean with poisoned data (i.e., $\epsilon(P; \alpha) - \epsilon(P|C; \alpha)$).

In all cases, the trend in the first plot is decreasing, which suggests that the gap increases in magnitude as the size of the incompatible data grows. Additionally, the clean label scenarios all exhibit the behavior that the source class accuracy increases as the amount of poison (and poisoned accuracy) increases, while the non-source accuracy drops. These attacks share the characteristic that the poisoned data are "harder" versions of the source class in the sense that they are perturbed toward instances of other classes. In the case of the GAN-based attack, the perturbations occur in the latent space of a GAN trained on clean data (e.g., the attacker identifies a bird and an airplane that are close in the latent space, linearly interpolates the latent representation of the airplane toward the latent representation of the bird, then uses the generative network to convert the perturbed representation into an image of a "bird-like airplane"). In the case of the $\ell_2$ and $\ell_\infty$ bounds, the attacker uses the Projected Gradient Descent (PGD) algorithm commonly used for adversarial training (Kurakin et al., 2016; Madry et al., 2017) to identify an image within a norm-bounded neighborhood of the original image that maximizes the loss of a trained network, i.e., the perturbed image is close in pixel-space to the original image, but was misclassified by the trained network as a different class. We therefore attribute the simultaneous increase in source class accuracy and decrease in non-source class accuracies to a similar mechanism for all the clean label cases.

Conversely, when measuring the compatibility of poisoned data with clean data (plots on the left) for the DLBD and WATERMARK attacks, we see that, as the amount of poisoned data increases, the source class accuracy drops. The implication is that the poisoned data causes the classifier to misclassify clean data by presenting similar (both quantitatively in terms of the distance between the poisoned and clean distributions, as well as perceptually in most settings) instances of clean source data that are mislabelled as the target class, thus complicating the learning of the clean source class.

Figures 10-14 plot the scenarios individually. We note that incompatibility holds generally across the range of scenarios, with only small deviations to the contrary.

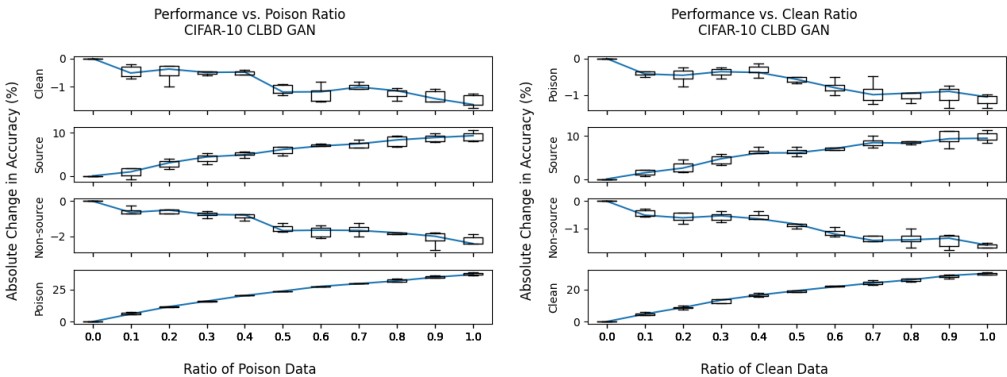

Figure 5: Aggregated compatibility results for the GAN-based CLBD attack on CIFAR-10.

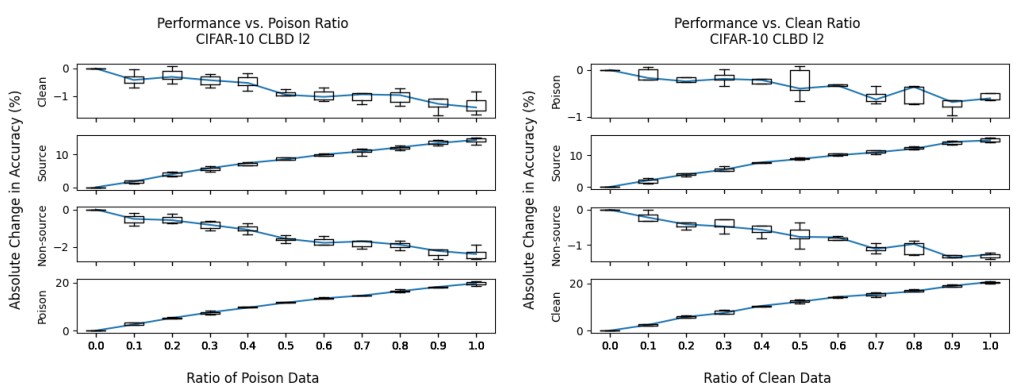

Figure 6: Aggregated compatibility results for the $\ell_2$-based CLBD attack on CIFAR-10.

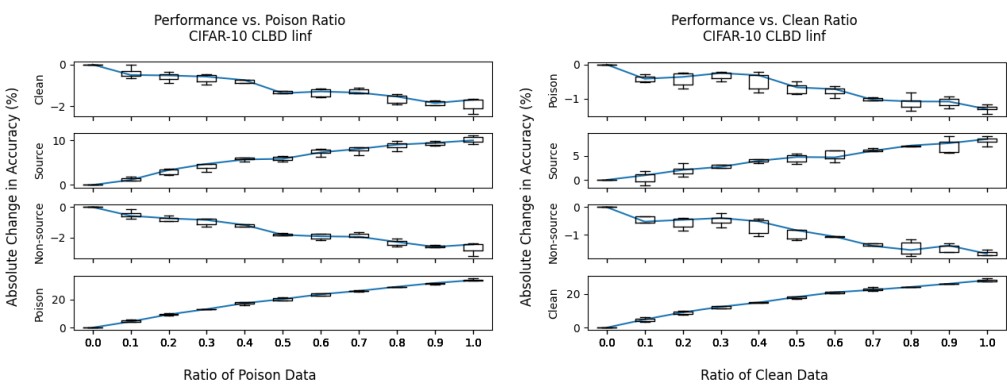

Figure 7: Aggregated compatibility results for the $\ell_\infty$-based CLBD attack on CIFAR-10.

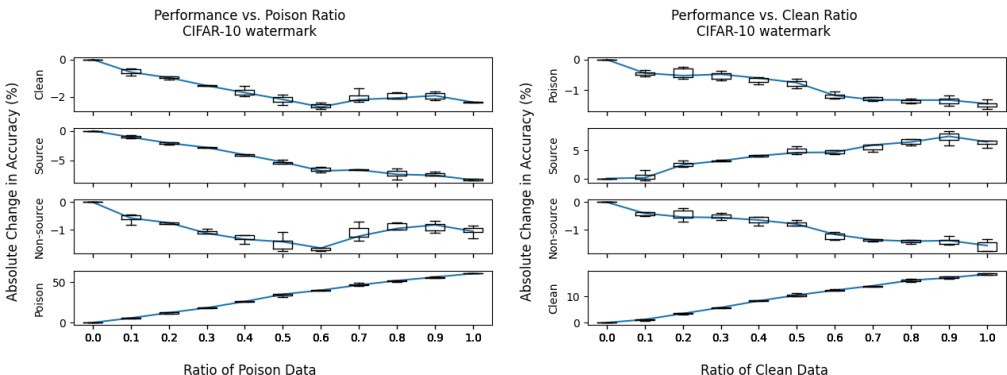

Figure 8: Aggregated compatibility results for the WATERMARK attack on CIFAR-10.

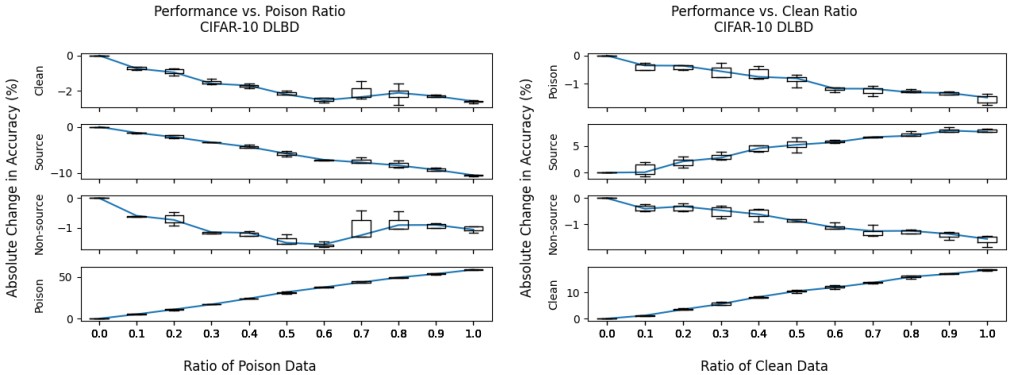

Figure 9: Aggregated compatibility results for the DLBD attack on CIFAR-10.

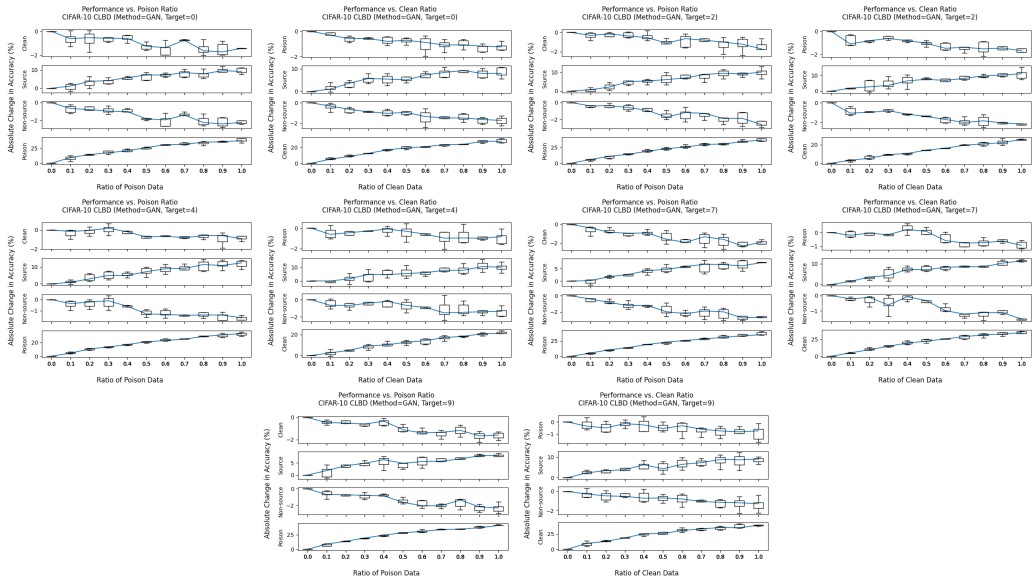

Figure 10: Full compatibility results for the GAN-based CLBD attack on CIFAR-10.

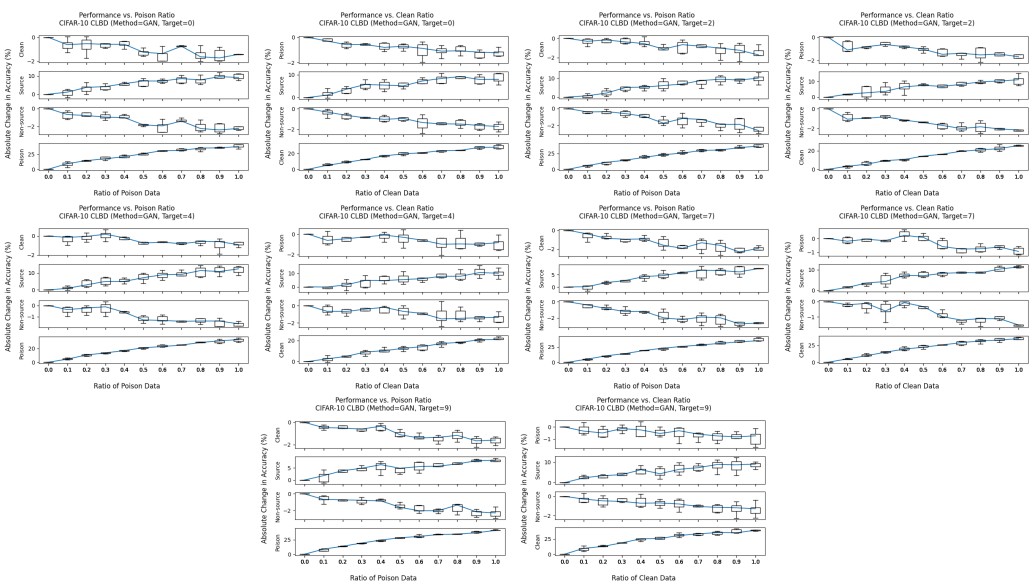

Figure 11: Full compatibility results for the $\ell_2$-based CLBD attack on CIFAR-10.

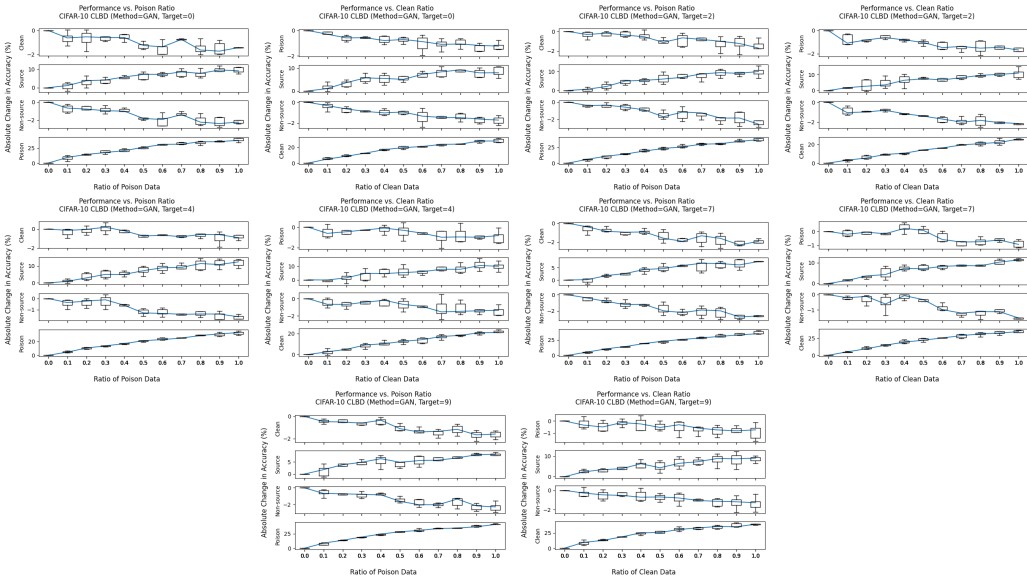

Figure 12: Full compatibility results for the $\ell_\infty$-based CLBD attack on CIFAR-10.

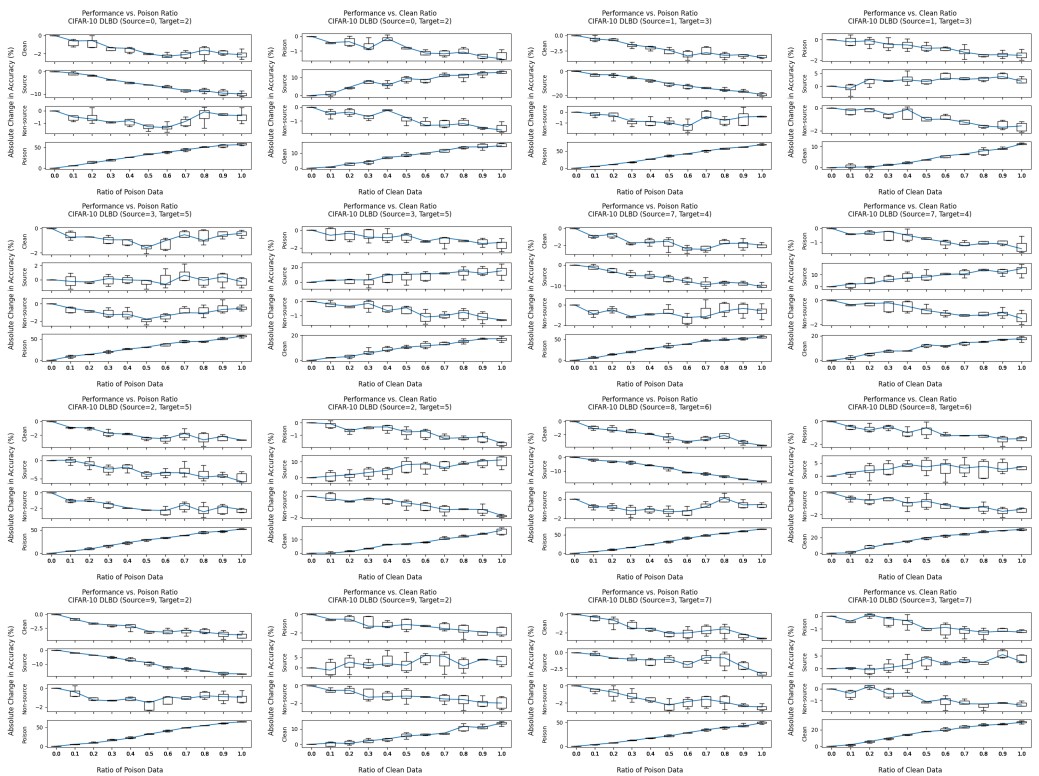

Figure 13: Full compatibility results for the DLBD attack on CIFAR-10.

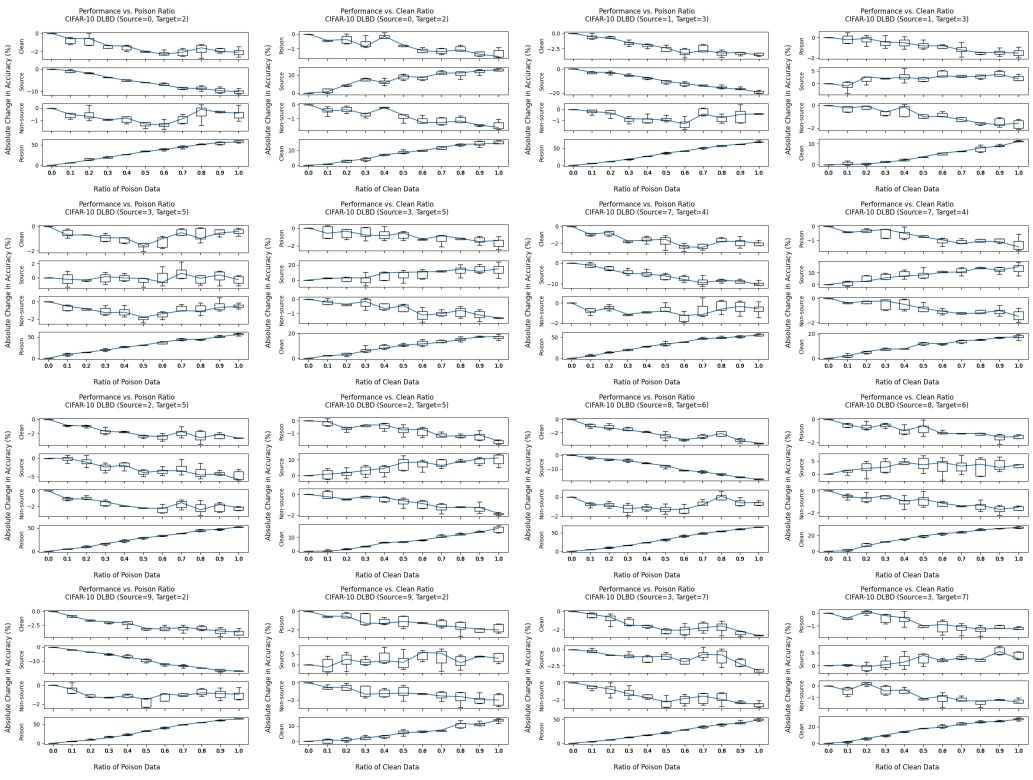

Figure 14: Full compatibility results for the WATERMARK attack on CIFAR-10.

### D.2 DEFENSE EVALUATION

This section includes additional experimental results concerning the performance of our defense. Tables 2-8 display the unabridged results of our main evaluation results for our defense (condensed in the main text as Table 1). We note that even though we have selected 1% as the threshold for a "successful" defense, in many of the failure cases, **ISPL+B** still achieves relatively low TMR. For instance, setting the threshold to 5% increases the number of successful scenarios to 142 (compared to 134 at 1%, out of 165 total).

Table 2 also includes results of the three baseline defenses for the standard 1-to-1 DLBD attack on CIFAR-10. In particular, we observe that our clean accuracy is comparable to (or even slightly better than) that of Activation Clustering (**AC**), the next best defense, despite **AC** achieving significantly lower TMR in our evaluations.

Table 2: Performance on the CIFAR-10 DLBD scenario (1-to-1) using the PreActResNet18 architecture. The S / T column lists the CIFAR-10 source and target classes. $\epsilon$ refers to the percentage of the source class which is poisoned. For the remainder of the columns, the top level column headers give the defense type: **Oracle** (training only on clean data), **ND** (no defense), **SS** (spectral signatures), **AC** (activation clustering), **ITLM** (iterative trimmed loss minimization, and **ISPL+B** (ISPL + boosting, this work); the second level column headers give the metric type: C (clean accuracy, higher is better), A (targeted misclassification rate, lower is better), FP (false positives, lower is better), FN (false negatives, lower is better). Successful runs (TMR < 1%) are highlighted.

| S / T | $\epsilon$ | Oracle C | Oracle A | ND C | ND A | SS C | SS A | AC C | AC A | ITLM C | ITLM A | ISPL+B C | ISPL+B A | ISPL+B FP | ISPL+B FN |
|---|---|---|---|---|---|---|---|---|---|---|---|---|---|---|---|
| 0 / 2 | 5 | 93.3 | 0.0 | 93.1 | 17.2 | 92.5 | 2.1 | 90.3 | 0.1 | 92.5 | 4.7 | 90.9 | 0.0 | 3761 | 23 |
|  | 10 | 93.1 | 0.0 | 92.8 | 42.6 | 91.3 | 18.2 | 90.7 | 14.7 | 92.5 | 43.7 | 90.6 | 0.0 | 3811 | 22 |
|  | 20 | 93.1 | 0.0 | 92.4 | 79.8 | 89.5 | 4.6 | 90.3 | 23.9 | 92.1 | 59.4 | 90.7 | 0.2 | 3773 | 118 |
| 1 / 3 | 5 | 93.1 | 0.0 | 92.7 | 1.4 | 92.4 | 0.3 | 90.6 | 0.3 | 92.6 | 0.0 | 90.9 | 0.0 | 3787 | 2 |
|  | 10 | 93.0 | 0.0 | 92.4 | 11.1 | 91.3 | 1.1 | 90.3 | 0.4 | 92.3 | 7.7 | 90.8 | 0.0 | 3758 | 5 |
|  | 20 | 93.1 | 0.0 | 92.5 | 55.7 | 88.2 | 4.3 | 90.0 | 4.8 | 92.2 | 51.6 | 90.6 | 0.0 | 3819 | 5 |
| 2 / 5 | 5 | 93.1 | 0.8 | 92.9 | 81.7 | 92.8 | 81.7 | 89.4 | 73.5 | 92.8 | 81.0 | 90.5 | 1.2 | 3778 | 44 |
|  | 10 | 93.3 | 0.9 | 92.9 | 83.3 | 92.1 | 81.0 | 90.4 | 76.2 | 92.5 | 82.1 | 90.6 | 0.6 | 3928 | 47 |
|  | 20 | 93.0 | 0.8 | 92.9 | 81.2 | 88.2 | 72.9 | 90.9 | 76.2 | 92.4 | 77.9 | 90.8 | 74.4 | 3552 | 981 |
| 3 / 5 | 5 | 93.2 | 0.2 | 92.8 | 72.9 | 92.6 | 69.9 | 91.0 | 56.5 | 92.8 | 68.0 | 90.8 | 0.6 | 3843 | 26 |
|  | 10 | 93.1 | 0.2 | 92.9 | 86.6 | 91.8 | 76.8 | 90.8 | 76.7 | 92.8 | 87.6 | 91.0 | 0.4 | 3759 | 42 |
|  | 20 | 93.1 | 0.2 | 92.8 | 89.3 | 90.2 | 81.7 | 90.4 | 80.6 | 92.7 | 88.9 | 90.4 | 0.2 | 3925 | 59 |
| 3 / 7 | 5 | 93.1 | 0.2 | 92.9 | 8.0 | 92.3 | 2.0 | 90.0 | 0.8 | 92.8 | 0.3 | 90.9 | 0.1 | 3656 | 12 |
|  | 10 | 92.9 | 0.0 | 92.8 | 46.6 | 91.3 | 20.8 | 89.7 | 3.4 | 92.4 | 36.5 | 90.7 | 0.1 | 3725 | 19 |
|  | 20 | 93.0 | 0.0 | 92.5 | 60.8 | 88.6 | 36.0 | 90.7 | 59.1 | 92.4 | 73.3 | 90.3 | 0.2 | 3895 | 100 |
| 7 / 4 | 5 | 93.0 | 0.0 | 93.2 | 76.2 | 92.5 | 58.2 | 90.7 | 15.6 | 92.6 | 40.6 | 91.0 | 0.0 | 3729 | 2 |
|  | 10 | 93.1 | 0.0 | 93.3 | 87.8 | 92.0 | 80.1 | 90.3 | 72.5 | 92.6 | 84.2 | 90.7 | 0.0 | 3781 | 3 |
|  | 20 | 93.2 | 0.0 | 93.1 | 92.9 | 88.8 | 36.3 | 90.6 | 74.0 | 92.6 | 91.1 | 90.8 | 0.0 | 3637 | 8 |
| 8 / 6 | 5 | 93.1 | 0.0 | 93.0 | 3.0 | 92.5 | 1.3 | 89.6 | 0.3 | 92.9 | 0.6 | 90.8 | 0.0 | 3818 | 2 |
|  | 10 | 93.3 | 0.0 | 93.1 | 87.8 | 91.2 | 47.0 | 91.0 | 2.9 | 92.3 | 15.3 | 90.6 | 0.0 | 4018 | 2 |
|  | 20 | 93.2 | 0.0 | 93.1 | 93.2 | 89.3 | 7.4 | 90.4 | 62.1 | 92.5 | 89.4 | 91.0 | 0.0 | 3697 | 4 |
| 9 / 2 | 5 | 93.3 | 0.0 | 92.9 | 75.4 | 92.7 | 14.5 | 90.3 | 31.0 | 92.7 | 1.0 | 91.0 | 0.0 | 3625 | 4 |
|  | 10 | 93.1 | 0.0 | 93.0 | 83.0 | 91.8 | 77.6 | 90.9 | 79.8 | 92.6 | 81.2 | 90.8 | 0.2 | 3718 | 37 |
|  | 20 | 93.0 | 0.0 | 92.3 | 41.2 | 89.4 | 76.6 | 91.1 | 78.5 | 92.6 | 81.2 | 90.6 | 0.2 | 3699 | 74 |

Table 3: Performance on the CIFAR-10 DLBD scenario (all-to-1) using the PreActResNet18 architecture. The T column lists the target class (i.e., all poisoned images have label T). Each of the 9 source classes contain $\epsilon/9$ percent poison.

| T | $\epsilon$ | Oracle | | ND | | ISPL+B | | | |
|---|---|---|---|---|---|---|---|---|---|
| | | C | A | C | A | C | A | FP | FN |
| | 5 | 93.2 | 0.0 | 93.0 | 0.1 | 91.1 | 0.0 | 3624 | 9 |
| 2 | 10 | 93.1 | 0.1 | 93.0 | 0.8 | 90.9 | 0.0 | 3632 | 25 |
| | 20 | 93.3 | 0.0 | 92.3 | 74.2 | 90.6 | 0.1 | 3770 | 49 |
| | 5 | 93.1 | 0.1 | 92.7 | 0.2 | 90.6 | 0.1 | 3921 | 8 |
| 3 | 10 | 93.0 | 0.1 | 92.4 | 25.4 | 90.6 | 0.1 | 3855 | 24 |
| | 20 | 93.3 | 0.1 | 92.8 | 87.6 | 90.3 | 0.1 | 3947 | 67 |
| | 5 | 93.2 | 0.2 | 93.0 | 62.8 | 90.8 | 0.1 | 3682 | 6 |
| 5 | 10 | 93.0 | 0.2 | 93.1 | 92.2 | 90.7 | 0.1 | 3764 | 21 |
| | 20 | 93.2 | 0.1 | 92.9 | 92.1 | 90.9 | 0.5 | 3698 | 66 |
| | 5 | 93.3 | 0.1 | 93.0 | 76.2 | 90.9 | 0.2 | 3605 | 8 |
| 7 | 10 | 93.0 | 0.1 | 92.8 | 83.3 | 90.8 | 0.2 | 3542 | 21 |
| | 20 | 93.0 | 0.1 | 92.8 | 89.1 | 90.9 | 0.5 | 3592 | 39 |

Table 4: Performance on the CIFAR-10 DLBD scenario (all-to-all) using the PreActResNet18 architecture. The S → T column lists offset of the cyclic permutation (i.e., 2 means that source class 0 is poisoned as target class 2, source class 1 is poisoned as target class 3, etc.). $\epsilon$ refers to the percentage of the source class which is poisoned.

| S → T | $\epsilon$ | Oracle | | ND | | ISPL+B | | | |
|---|---|---|---|---|---|---|---|---|---|
| | | C | A | C | A | C | A | FP | FN |
| | 5 | 92.7 | 0.0 | 92.4 | 78.0 | 90.3 | 0.1 | 3790 | 130 |
| 2 | 10 | 92.8 | 0.0 | 92.5 | 87.8 | 90.0 | 0.1 | 3648 | 256 |
| | 20 | 92.2 | 0.1 | 92.6 | 88.2 | 88.2 | 0.2 | 3695 | 821 |
| | 5 | 92.8 | 0.0 | 93.2 | 88.2 | 90.6 | 0.0 | 3664 | 58 |
| 3 | 10 | 92.5 | 0.0 | 92.9 | 89.7 | 89.6 | 0.0 | 3892 | 172 |
| | 20 | 92.2 | 0.0 | 92.6 | 90.7 | 88.4 | 0.1 | 3902 | 384 |
| | 5 | 93.1 | 0.0 | 93.0 | 90.1 | 90.7 | 0.1 | 3446 | 57 |
| 5 | 10 | 92.8 | 0.0 | 93.2 | 90.6 | 90.4 | 8.9 | 3416 | 410 |
| | 20 | 92.4 | 0.1 | 92.9 | 91.0 | 90.3 | 88.5 | 3367 | 9027 |
| | 5 | 93.1 | 0.1 | 93.1 | 85.6 | 90.6 | 0.1 | 3505 | 58 |
| 7 | 10 | 92.7 | 0.0 | 93.2 | 89.0 | 90.6 | 69.4 | 3532 | 3576 |
| | 20 | 92.5 | 0.0 | 93.2 | 90.0 | 90.6 | 86.7 | 3076 | 8952 |

Table 5: Performance on the CIFAR-10 WATERMARK scenario (1-to-1) using the PreActResNet18 architecture. The S / T column lists the CIFAR-10 source and target classes. $\epsilon$ refers to the percentage of the source class which is poisoned.

| S / T | $\epsilon$ | Oracle | | ND | | ISPL+B | | | |
|---|---|---|---|---|---|---|---|---|---|
| | | C | A | C | A | C | A | FP | FN |
| 0 / 2 | 5 | 93.1 | 0.2 | 93.3 | 75.2 | 90.8 | 0.4 | 3826 | 13 |
| | 10 | 93.2 | 0.3 | 93.2 | 84.5 | 90.6 | 4.4 | 3868 | 54 |
| | 20 | 93.1 | 0.1 | 93.0 | 87.0 | 90.6 | 76.8 | 3741 | 787 |
| 1 / 3 | 5 | 93.1 | 0.0 | 93.0 | 47.7 | 90.8 | 0.0 | 3752 | 2 |
| | 10 | 93.0 | 0.0 | 93.1 | 74.4 | 90.8 | 0.0 | 3746 | 5 |
| | 20 | 93.2 | 0.0 | 92.9 | 85.8 | 90.9 | 0.0 | 3634 | 11 |
| 2 / 5 | 5 | 93.0 | 2.0 | 93.0 | 45.2 | 90.7 | 11.9 | 3685 | 109 |
| | 10 | 93.1 | 2.4 | 92.9 | 64.4 | 90.7 | 7.1 | 3664 | 122 |
| | 20 | 93.1 | 2.4 | 92.9 | 73.7 | 90.4 | 35.4 | 3499 | 622 |
| 3 / 5 | 5 | 93.0 | 0.1 | 93.0 | 45.1 | 90.6 | 0.2 | 3814 | 35 |
| | 10 | 93.1 | 0.1 | 93.0 | 74.3 | 90.8 | 0.3 | 3715 | 44 |
| | 20 | 92.9 | 0.5 | 92.9 | 85.1 | 90.4 | 21.7 | 3796 | 289 |
| 3 / 7 | 5 | 93.2 | 0.7 | 92.9 | 84.2 | 90.7 | 6.5 | 3920 | 16 |
| | 10 | 93.1 | 0.9 | 92.9 | 85.9 | 90.7 | 73.5 | 3717 | 406 |
| | 20 | 93.3 | 1.2 | 92.9 | 86.1 | 91.0 | 74.9 | 3772 | 928 |
| 7 / 4 | 5 | 92.9 | 0.0 | 93.1 | 77.6 | 90.8 | 0.0 | 3877 | 4 |
| | 10 | 92.9 | 0.0 | 92.9 | 82.9 | 90.7 | 0.0 | 3756 | 5 |
| | 20 | 93.0 | 0.0 | 92.9 | 89.4 | 90.6 | 72.4 | 3907 | 711 |
| 8 / 6 | 5 | 93.1 | 0.0 | 93.1 | 82.4 | 90.7 | 0.0 | 3639 | 1 |
| | 10 | 93.2 | 0.0 | 93.2 | 87.7 | 90.7 | 0.1 | 3751 | 3 |
| | 20 | 93.1 | 0.0 | 93.2 | 92.5 | 90.5 | 83.2 | 3795 | 859 |
| 9 / 2 | 5 | 93.1 | 0.3 | 93.1 | 69.6 | 91.0 | 0.8 | 3634 | 18 |
| | 10 | 92.9 | 0.1 | 92.9 | 75.0 | 90.7 | 0.4 | 3707 | 17 |
| | 20 | 92.9 | 0.2 | 92.8 | 78.8 | 90.6 | 1.3 | 3519 | 58 |

Table 6: Performance on the CIFAR-10 CLBD scenario using the PreActResNet18 architecture. The T column lists the target class (i.e., all poisoned images have label T). $\epsilon$ refers to the percentage of the target class which is poisoned.

| M | T | $\epsilon$ | Oracle C | Oracle A | ND C | ND A | ISPL+B C | ISPL+B A | FP | FN |
|---|---|---|---|---|---|---|---|---|---|---|
| GAN | 0 | 5 | 93.2 | 0.0 | 93.2 | 0.1 | 90.9 | 0.1 | 3654 | 192 |
| | | 10 | 93.1 | 0.0 | 93.0 | 0.1 | 90.9 | 0.1 | 3763 | 406 |
| | | 20 | 93.1 | 0.0 | 93.3 | 0.3 | 91.0 | 0.2 | 3688 | 968 |
| | 2 | 5 | 93.0 | 0.1 | 93.2 | 1.3 | 90.7 | 0.2 | 3635 | 129 |
| | | 10 | 92.9 | 0.1 | 92.9 | 12.1 | 90.7 | 0.4 | 3739 | 324 |
| | | 20 | 93.1 | 0.1 | 93.3 | 27.2 | 90.8 | 4.3 | 3621 | 907 |
| | 4 | 5 | 92.9 | 0.1 | 93.3 | 0.2 | 90.9 | 0.1 | 3666 | 204 |
| | | 10 | 92.9 | 0.1 | 93.0 | 0.4 | 90.8 | 0.1 | 3735 | 406 |
| | | 20 | 92.9 | 0.0 | 93.1 | 3.4 | 91.0 | 0.3 | 3532 | 875 |
| | 7 | 5 | 93.1 | 0.0 | 93.2 | 0.5 | 90.9 | 0.0 | 3763 | 124 |
| | | 10 | 93.3 | 0.0 | 93.3 | 4.8 | 90.6 | 0.1 | 3832 | 217 |
| | | 20 | 92.9 | 0.0 | 93.0 | 25.2 | 90.8 | 0.2 | 3595 | 574 |
| | 9 | 5 | 93.2 | 0.0 | 93.0 | 0.3 | 91.0 | 0.1 | 3742 | 143 |
| | | 10 | 93.1 | 0.1 | 93.1 | 4.7 | 90.9 | 0.1 | 3663 | 400 |
| | | 20 | 93.1 | 0.0 | 93.2 | 10.6 | 90.7 | 0.7 | 3943 | 913 |
| $\ell_2$ | 0 | 5 | 93.0 | 0.0 | 93.2 | 0.3 | 90.8 | 0.0 | 3633 | 162 |
| | | 10 | 93.2 | 0.0 | 93.1 | 6.1 | 90.6 | 0.1 | 3747 | 351 |
| | | 20 | 92.9 | 0.0 | 93.2 | 45.8 | 90.6 | 1.1 | 3848 | 757 |
| | 2 | 5 | 92.9 | 0.0 | 93.2 | 30.1 | 91.0 | 0.2 | 3694 | 95 |
| | | 10 | 93.2 | 0.1 | 93.1 | 68.9 | 90.4 | 0.2 | 3877 | 219 |
| | | 20 | 92.9 | 0.1 | 93.1 | 76.4 | 90.6 | 20.3 | 3716 | 633 |
| | 4 | 5 | 93.0 | 0.1 | 93.5 | 6.8 | 90.9 | 0.1 | 3639 | 140 |
| | | 10 | 93.0 | 0.1 | 93.0 | 57.4 | 90.6 | 0.5 | 3793 | 299 |
| | | 20 | 93.0 | 0.0 | 93.1 | 74.4 | 90.2 | 4.8 | 3731 | 587 |
| | 7 | 5 | 92.9 | 0.0 | 93.0 | 0.3 | 91.0 | 0.1 | 3769 | 157 |
| | | 10 | 93.3 | 0.0 | 92.9 | 22.1 | 90.7 | 0.1 | 3796 | 366 |
| | | 20 | 92.9 | 0.0 | 93.1 | 68.0 | 90.5 | 0.4 | 3932 | 732 |
| | 9 | 5 | 93.1 | 0.1 | 93.1 | 0.4 | 91.0 | 0.1 | 3871 | 174 |
| | | 10 | 93.4 | 0.1 | 93.2 | 12.5 | 90.6 | 0.1 | 3846 | 332 |
| | | 20 | 92.9 | 0.0 | 93.1 | 68.6 | 90.9 | 0.9 | 3622 | 731 |
| $\ell_\infty$ | 0 | 5 | 93.2 | 0.0 | 93.1 | 0.5 | 91.1 | 0.0 | 3732 | 109 |
| | | 10 | 93.1 | 0.0 | 93.0 | 3.8 | 91.1 | 0.1 | 3591 | 301 |
| | | 20 | 93.2 | 0.0 | 93.1 | 51.3 | 90.7 | 0.2 | 3686 | 729 |
| | 2 | 5 | 93.1 | 0.1 | 93.2 | 13.7 | 90.8 | 0.1 | 3723 | 44 |
| | | 10 | 93.3 | 0.1 | 93.3 | 52.9 | 90.9 | 0.2 | 3641 | 120 |
| | | 20 | 93.0 | 0.1 | 93.0 | 66.2 | 90.4 | 1.0 | 4157 | 430 |
| | 4 | 5 | 93.1 | 0.1 | 93.2 | 0.6 | 90.7 | 0.1 | 3919 | 67 |
| | | 10 | 93.2 | 0.1 | 93.1 | 14.4 | 90.8 | 0.1 | 3918 | 183 |
| | | 20 | 93.1 | 0.0 | 93.1 | 4.1 | 90.6 | 0.2 | 3807 | 716 |
| | 7 | 5 | 93.2 | 0.0 | 93.0 | 1.0 | 90.6 | 0.0 | 3592 | 97 |
| | | 10 | 93.2 | 0.0 | 93.0 | 32.6 | 90.6 | 0.1 | 3780 | 235 |
| | | 20 | 93.0 | 0.0 | 93.1 | 67.4 | 90.5 | 0.2 | 3625 | 492 |
| | 9 | 5 | 93.1 | 0.1 | 93.1 | 0.7 | 91.0 | 0.1 | 3683 | 109 |
| | | 10 | 93.1 | 0.0 | 93.1 | 3.7 | 90.6 | 0.2 | 3929 | 308 |
| | | 20 | 93.2 | 0.1 | 93.4 | 1.0 | 90.7 | 0.2 | 3697 | 792 |

Table 7: Performance on the GTSRB DLBD scenario (1-to-1) using the ResNet32 architecture. The S / T column lists the GTSRB source and target classes. $\epsilon$ refers to the percentage of the source class which is poisoned.

| S / T | $\epsilon$ | Oracle | | ND | | ISPL+B | | | |
|---|---|---|---|---|---|---|---|---|---|
| | | C | A | C | A | C | A | FP | FN |
| 0 / 2 | 5 | 94.2 | 0.0 | 94.4 | 0.0 | 95.1 | 0.0 | 628 | 0 |
| | 10 | 94.9 | 0.0 | 94.2 | 0.0 | 94.8 | 0.0 | 591 | 0 |
| | 20 | 94.4 | 0.0 | 94.4 | 0.0 | 94.1 | 0.0 | 558 | 1 |
| 1 / 2 | 5 | 93.9 | 0.0 | 94.5 | 0.1 | 94.4 | 0.0 | 327 | 1 |
| | 10 | 94.3 | 0.0 | 94.5 | 0.1 | 94.3 | 0.0 | 430 | 8 |
| | 20 | 94.2 | 0.0 | 93.7 | 53.2 | 93.7 | 0.1 | 418 | 25 |
| 1 / 0 | 5 | 94.5 | 0.0 | 94.1 | 87.5 | 94.3 | 0.0 | 582 | 0 |
| | 10 | 94.0 | 0.0 | 94.6 | 97.6 | 94.5 | 0.0 | 460 | 6 |
| | 20 | 94.2 | 0.0 | 94.5 | 97.9 | 93.0 | 82.6 | 569 | 79 |
| 2 / 13 | 5 | 94.2 | 0.0 | 94.5 | 91.9 | 94.2 | 89.7 | 269 | 75 |
| | 10 | 94.3 | 0.0 | 94.8 | 97.5 | 94.7 | 95.7 | 366 | 150 |
| | 20 | 94.2 | 0.0 | 93.8 | 96.8 | 93.9 | 98.1 | 472 | 300 |
| 38 / 4 | 5 | 94.5 | 0.0 | 94.1 | 1.6 | 94.2 | 4.9 | 453 | 69 |
| | 10 | 94.1 | 0.0 | 92.7 | 3.0 | 93.8 | 2.3 | 668 | 138 |
| | 20 | 94.6 | 0.0 | 94.1 | 9.3 | 93.7 | 30.1 | 665 | 276 |
| 31 / 24 | 5 | 94.3 | 0.0 | 94.0 | 47.4 | 95.2 | 0.0 | 616 | 0 |
| | 10 | 94.2 | 0.0 | 93.5 | 60.7 | 93.8 | 0.0 | 553 | 0 |
| | 20 | 93.9 | 0.0 | 93.9 | 83.0 | 94.7 | 0.0 | 643 | 7 |
| 15 / 29 | 5 | 94.5 | 0.0 | 94.5 | 0.5 | 94.1 | 0.0 | 674 | 7 |
| | 10 | 94.3 | 0.0 | 94.4 | 0.5 | 94.5 | 0.0 | 649 | 10 |
| | 20 | 94.1 | 0.0 | 94.4 | 0.0 | 94.6 | 0.5 | 706 | 74 |
| 9 / 18 | 5 | 92.8 | 0.0 | 93.8 | 61.9 | 94.3 | 0.0 | 586 | 2 |
| | 10 | 93.8 | 0.0 | 94.6 | 96.9 | 94.4 | 0.0 | 607 | 2 |
| | 20 | 94.3 | 0.0 | 94.2 | 98.8 | 94.3 | 0.0 | 586 | 26 |

Table 8: Performance on the GTSRB WATERMARK scenario (1-to-1) using the ResNet32 architecture. The S / T column lists the GTSRB source and target classes. $\epsilon$ refers to the percentage of the source class which is poisoned.

| S / T | $\epsilon$ | Oracle | | ND | | ISPL+B | | | |
|-------|-----------|--------|-----|-----|------|--------|------|-----|-----|
| | | C | A | C | A | C | A | FP | FN |
| 0 / 2 | 5 | 94.6 | 0.0 | 94.5 | 0.0 | 93.0 | 0.0 | 369 | 0 |
| | 10 | 94.4 | 0.0 | 93.8 | 0.0 | 94.0 | 0.0 | 416 | 0 |
| | 20 | 94.1 | 0.0 | 94.1 | 0.0 | 93.7 | 0.0 | 783 | 4 |
| 1 / 2 | 5 | 94.4 | 0.0 | 94.5 | 10.3 | 94.2 | 0.0 | 452 | 1 |
| | 10 | 94.1 | 0.0 | 94.2 | 54.0 | 94.7 | 0.0 | 359 | 8 |
| | 20 | 94.5 | 0.0 | 94.1 | 77.5 | 94.2 | 0.0 | 747 | 17 |
| 1 / 0 | 5 | 94.2 | 0.0 | 94.1 | 0.6 | 93.7 | 0.0 | 379 | 0 |
| | 10 | 94.9 | 0.0 | 94.2 | 10.7 | 94.3 | 0.0 | 305 | 5 |
| | 20 | 94.9 | 0.0 | 94.5 | 73.3 | 93.1 | 0.6 | 360 | 80 |
| 2 / 13 | 5 | 94.5 | 0.0 | 94.1 | 1.6 | 94.7 | 0.0 | 270 | 0 |
| | 10 | 93.7 | 0.0 | 93.9 | 32.1 | 94.8 | 0.0 | 278 | 0 |
| | 20 | 94.4 | 0.0 | 94.0 | 76.9 | 94.2 | 0.1 | 401 | 300 |
| 38 / 4 | 5 | 94.4 | 0.0 | 94.2 | 50.0 | 93.9 | 0.0 | 308 | 0 |
| | 10 | 94.2 | 0.0 | 93.3 | 65.4 | 94.3 | 0.0 | 273 | 0 |
| | 20 | 94.2 | 0.0 | 94.2 | 78.1 | 94.0 | 66.4 | 685 | 275 |
| 31 / 24 | 5 | 94.4 | 0.0 | 94.5 | 69.3 | 94.0 | 0.0 | 378 | 0 |
| | 10 | 94.6 | 0.0 | 94.7 | 81.1 | 94.3 | 0.0 | 292 | 0 |
| | 20 | 94.8 | 0.0 | 94.5 | 80.4 | 94.4 | 0.0 | 566 | 8 |
| 15 / 29 | 5 | 94.4 | 0.0 | 94.5 | 1.0 | 94.2 | 0.0 | 302 | 0 |
| | 10 | 94.5 | 0.0 | 94.1 | 0.5 | 94.6 | 0.0 | 344 | 0 |
| | 20 | 94.4 | 0.0 | 94.5 | 47.6 | 94.4 | 19.0 | 482 | 59 |
| 9 / 18 | 5 | 93.9 | 0.0 | 93.8 | 6.9 | 94.3 | 0.0 | 331 | 0 |
| | 10 | 94.3 | 0.0 | 94.3 | 75.0 | 94.3 | 0.0 | 269 | 0 |
| | 20 | 94.8 | 0.0 | 94.6 | 94.0 | 94.2 | 0.0 | 482 | 0 |

### D.2.1 SLEEPER AGENT ATTACK

Sleeper Agent (Souri et al., 2022) is a clean label attack that uses gradient matching (Geiping et al., 2020) to create a poisoned training set. The objective of the attacker is to induce a model trained on the poisoned training set to misclassify patched versions of a source class as the target class. Figure 16 displays examples of clean and poisoned training and test data created by the Sleeper Agent attack, where the source class is horse and the target class is dog (training with the poisoned dogs causes patched horses to be misclassified as dogs). Similar to the CLBD attack, Sleeper Agent perturbs each image in the training set independently subject to an $\ell_\infty$ bound, however, the Sleeper Agent attack is more subtle because it does not insert any patches during training. The Sleeper Agent attacker also differs from the DLBD, CLBD, and WATERMARK attacks in using an informed mechanism to select which subset of the training set $D$ to poison (rather than poisoning a random subset). Finally, unlike prior gradient matching approaches, the Sleeper Agent attack is also intended to apply in the black-box setting, where the attacker does not have access to the victim model's architecture or training hyperparameters.

We first used the official repository released by the authors to generate 24 poisoned datasets with the default settings for CIFAR-10 using different random seeds, targeting a ResNet32 as the victim architecture.[1] We observed an attack success rate (ASR) of 49.1% ($\pm$38.3%) over the generated datasets, when training a ResNet32 model with the attacker's original training hyperparameters. We then transferred the poisoned dataset to the setting of our defense (training a ResNet32, using the same training and defense hyperparameters as in our main experiments for ResNet32 on CIFAR-10). Our defense delivers an ASR of 4.3% ($\pm$4.1%), compared to an oracle, which achieves an ASR of 3.0% ($\pm$4.8%). However, we also observed that the attack only achieves a success rate of 4.5% ($\pm$4.3%) on an *undefended* model in our setting. We attribute this anomaly (between the ASR for an undefended model in our setting and the ASR reported by the attacker) to the difference in training hyperparameters, despite using the same victim architecture. In all three cases—defended, undefended, and oracle—the clean accuracy is consistent between 89-90%.

We next evaluated the effectiveness of our defense when retraining with the attacker's training setup. Specifically, we took the poisoned training set generated by the attacker, then ran our defense using our training setup to output a new training set with the detected poisoned examples removed. We then used the attacker's training setup to train a new model on our new training set, and evaluated this new model on the poisoned test set.

Note that in our threat model, the training algorithm $\mathcal{A}$ is fixed, and our defense is designed around the interaction between the data and $\mathcal{A}$. The above scenario is thus explicitly not the intended use case for our algorithm. Nevertheless, our defense still achieves an ASR of 14.1% ($\pm$15.6%), with a clean accuracy of 87.5% ($\pm$0.6%), compared to the original ASR of 49.1%. An oracle achieves an ASR of 2.4% ($\pm$2.2). These results thus indicate that our defense successfully defends against the Sleeper Agent attack in our threat model (where the defense is integrated in the intended training pipeline), while also remaining robust to an evaluation under the attacker's original training setup.

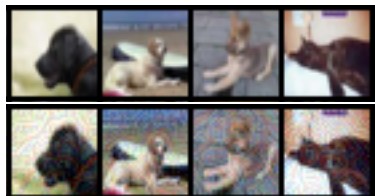 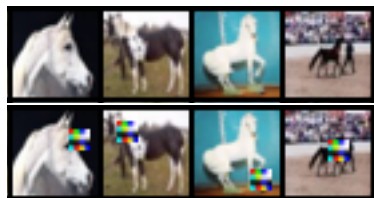

Figure 15: Example clean (top) and poisoned (bottom) images from the train (left) and test (right) datasets, taken from one run of Sleeper Agent attack; the source class is horse, and the target class is dog (training with the poisoned dogs causes patched horses to be misclassified as dogs).

---

[1] All results in this section are the mean over 24 runs, with the standard error in parentheses, to match the presentation in Souri et al. (2022) .

### D.2.2 IMAGENETTE DATASET

Imagenette (Howard, 2019) is a subset the ImageNet dataset (Deng et al., 2009) consisting of 9,469 training and 3,925 test images over 10 classes. We downsampled the 320x320 resolution version of Imagenette to 224x224, and use a ResNet56 achieving around 91% accuracy on the clean dataset. We used a DLBD adversary that places a solid 5x5 patch of a random color in a random position on the image. We use this setting to probe the applicability of our technique to higher-resolution datasets. Figure 16 displays examples of clean and poisoned data from the Imagenette experiments. Table 9 reports the results of our defense on Imagenette, using the same defense hyperparameters as the main experiments for CIFAR-10. The performance of our defense remains consistent, with 21 of the 24 scenarios defended below 1% TMR and less than 2% drop in clean accuracy.

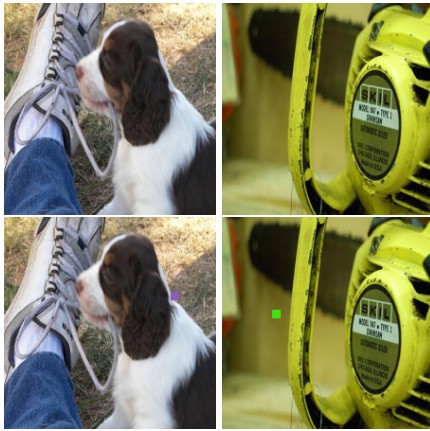

Figure 16: Examples of clean (top) and poisoned (bottom) data from the Imagenette experiments.

Table 9: Performance on the Imagenette DLBD scenario (1-to-1) using the ResNet56 architecture. The S / T column lists the Imagenette source and target classes. $\epsilon$ refers to the percentage of the source class which is poisoned.

| S / T | $\epsilon$ | Oracle | | ND | | ISPL+B | | | |
|-------|-----|------|-----|------|------|------|------|-----|-----|
| | | C | A | C | A | C | A | FP | FN |
| 0 / 2 | 5 | 90.9 | 0.0 | 91.2 | 85.5 | 89.3 | 0.0 | 544 | 0 |
| | 10 | 91.3 | 0.0 | 91.3 | 93.8 | 89.5 | 0.0 | 535 | 2 |
| | 20 | 91.2 | 0.0 | 91.0 | 92.2 | 88.7 | 0.3 | 608 | 17 |
| 1 / 3 | 5 | 91.1 | 0.0 | 91.3 | 79.7 | 89.1 | 0.0 | 536 | 10 |
| | 10 | 90.7 | 0.0 | 91.0 | 87.1 | 89.2 | 0.5 | 513 | 19 |
| | 20 | 91.4 | 0.0 | 91.2 | 91.9 | 88.9 | 89.1 | 586 | 185 |
| 2 / 5 | 5 | 91.1 | 0.0 | 91.0 | 0.5 | 88.7 | 0.0 | 619 | 7 |
| | 10 | 91.5 | 0.0 | 90.8 | 72.8 | 89.1 | 0.0 | 642 | 23 |
| | 20 | 90.9 | 0.3 | 90.7 | 77.2 | 88.3 | 0.8 | 639 | 65 |
| 3 / 5 | 5 | 91.3 | 0.0 | 91.3 | 69.7 | 89.5 | 0.0 | 554 | 6 |
| | 10 | 90.8 | 0.0 | 91.0 | 79.0 | 88.7 | 0.0 | 599 | 13 |
| | 20 | 91.2 | 0.0 | 90.9 | 79.7 | 88.5 | 72.8 | 572 | 182 |
| 3 / 7 | 5 | 90.6 | 0.0 | 90.3 | 13.2 | 89.9 | 0.0 | 536 | 2 |
| | 10 | 91.5 | 0.0 | 91.0 | 88.2 | 89.4 | 0.0 | 520 | 3 |
| | 20 | 91.7 | 0.0 | 91.1 | 90.8 | 88.9 | 0.0 | 617 | 24 |
| 7 / 4 | 5 | 91.7 | 0.0 | 90.7 | 72.2 | 89.7 | 0.0 | 562 | 0 |
| | 10 | 91.3 | 0.0 | 91.3 | 81.7 | 89.2 | 0.0 | 521 | 2 |
| | 20 | 91.4 | 0.0 | 90.6 | 86.2 | 89.8 | 0.0 | 542 | 10 |
| 8 / 6 | 5 | 90.7 | 0.0 | 91.3 | 83.3 | 89.8 | 0.0 | 540 | 1 |
| | 10 | 91.5 | 0.0 | 91.2 | 91.3 | 89.1 | 0.0 | 569 | 5 |
| | 20 | 90.9 | 0.0 | 91.5 | 92.1 | 89.1 | 0.0 | 495 | 5 |
| 9 / 2 | 5 | 91.7 | 0.0 | 90.9 | 17.9 | 89.6 | 0.0 | 496 | 3 |
| | 10 | 91.0 | 0.0 | 90.9 | 79.8 | 89.4 | 0.0 | 544 | 4 |
| | 20 | 91.4 | 0.0 | 90.7 | 76.2 | 88.5 | 2.3 | 581 | 58 |

### D.2.3 ADAPTIVE ATTACKER

The attacks used in the main experiments are oblivious to the presence of a defender when selecting the data to poison. In this section, we devise an strong adaptive attacker with white-box access to the ISPL algorithm when constructing the poisoned dataset.

The success of the defense rests crucial on the partition of training data returned by the ISPL algorithm. In order for the boosting phase to succeed, the partition should consist of homogeneous sets, i.e., the poison must be concentrated in a small number of sets within the partition; hence, the adaptive attacker's objective is to try and distribute the poisoned samples across multiple sets.

In particular, rather than randomly selecting a $\epsilon$ fraction of the source class to poison, an adaptive attacker with white-box access to the defense can first run the ISPL algorithm (using the same parameters) on the clean dataset to create a partition of the dataset, then random select a $\epsilon$ fraction *from each partition* to poison. The idea is that if the ISPL algorithm returns a similar partition, then the poison will be still be distributed across multiple sets, thereby breaking the defense.

Table 10 presents the results of our defense against the strong adaptive attacker described above. Against the adaptive attack, our defense still succeeds in 20/24 of the scenarios (versus 22/24 for the standard oblivious attack). These results suggest the ISPL algorithm remains robust even to white-box adaptive attacks.

Table 10: Performance on the CIFAR-10 DLBD scenario using the PreActResNet18 architecture with an adaptive attack. The S / T column lists the CIFAR-10 source and target classes. $\epsilon$ refers to the percentage of the source class which is poisoned.

| S / T | $\epsilon$ | ISPL+B | | | |
|---|---|---|---|---|---|
| | | C | A | FP | FN |
| | 5 | 90.6 | 0.0 | 3899 | 23 |
| 0 / 2 | 10 | 91.0 | 0.0 | 3594 | 17 |
| | 20 | 90.6 | 0.2 | 3819 | 78 |
| | 5 | 90.9 | 0.0 | 3733 | 1 |
| 1 / 3 | 10 | 91.0 | 0.0 | 3796 | 4 |
| | 20 | 90.8 | 0.0 | 3797 | 17 |
| | 5 | 90.7 | 0.9 | 3741 | 46 |
| 2 / 5 | 10 | 90.8 | 64.5 | 3656 | 336 |
| | 20 | 90.6 | 74.0 | 3558 | 953 |
| | 5 | 90.8 | 0.3 | 3562 | 10 |
| 3 / 5 | 10 | 90.7 | 8.2 | 3665 | 51 |
| | 20 | 90.4 | 69.4 | 4162 | 731 |
| | 5 | 90.6 | 0.1 | 3665 | 6 |
| 3 / 7 | 10 | 90.6 | 0.1 | 3754 | 23 |
| | 20 | 90.6 | 0.3 | 3526 | 46 |
| | 5 | 90.8 | 0.0 | 3746 | 1 |
| 7 / 4 | 10 | 91.0 | 0.0 | 3730 | 2 |
| | 20 | 90.8 | 0.0 | 3559 | 11 |
| | 5 | 90.7 | 0.0 | 3760 | 2 |
| 8 / 6 | 10 | 90.9 | 0.0 | 3723 | 5 |
| | 20 | 90.7 | 0.0 | 3818 | 4 |
| | 5 | 90.8 | 0.0 | 3738 | 11 |
| 9 / 2 | 10 | 90.9 | 0.1 | 3660 | 21 |
| | 20 | 90.8 | 0.4 | 3640 | 70 |

### D.2.4 VARYING THE TRAINING SETUP

As prior works (Schwarzschild et al., 2021) have observed that results in data poisoning, particularly with DNNs, can vary significantly with the training setup, we additionally ran our defense (along with the three baseline defenses) on the standard CIFAR-10 DLBD attack using both the PreActResNet18 and ResNet32 models, respectively, and report results after training for 200 epochs (instead of 100, as in our main experiments).

Tables 11 and 12 report the results of these experiments. We observe that the defense success rates are largely consistent with those in the main experiments across all defenses; the biggest change is that the Spectral Signatures defense successfully defends against 9 scenarios instead of just 1 (out of 24 total). **ISPL+B** still achieves the best performance in terms of reducing the targeted misclassification rate below 1% (with a 2-3% drop in clean accuracy).

Table 11: Performance on the CIFAR-10 DLBD scenario (1-to-1) using the PreActResNet18 architecture, when training for the final model for 200 epochs. The S / T column lists the CIFAR-10 source and target classes. $\epsilon$ refers to the percentage of the source class which is poisoned.

| S / T | $\epsilon$ | Oracle A | ND C | ND A | SS C | SS A | SS FP | SS FN | AC C | AC A | AC FP | AC FN | ITLM C | ITLM A | ITLM FP | ITLM FN | ISPL+B C | ISPL+B A | ISPL+B FP | ISPL+B FN |
|---|---|---|---|---|---|---|---|---|---|---|---|---|---|---|---|---|---|---|---|---|
| 0 / 2 | 5 | 1.3 | 94.5 | 91.3 | 94.5 | 79.9 | 7381 | 130 | 91.9 | 58.3 | 18926 | 155 | 94.6 | 84.9 | 994 | 244 | 92.8 | 0.0 | 3640 | 23 |
|  | 10 | 1.3 | 94.1 | 90.6 | 94.5 | 66.0 | 7383 | 383 | 92.3 | 85.9 | 19790 | 320 | 94.2 | 92.8 | 1961 | 461 | 92.8 | 0.0 | 3479 | 25 |
|  | 20 | 1.1 | 94.6 | 80.2 | 94.2 | 64.2 | 14335 | 335 | 92.4 | 73.9 | 19127 | 601 | 93.8 | 93.8 | 3897 | 897 | 91.6 | 22.9 | 3711 | 237 |
| 1 / 3 | 5 | 0.0 | 94.4 | 92.9 | 94.7 | 5.5 | 7319 | 68 | 90.9 | 19.5 | 19791 | 155 | 94.8 | 96.5 | 986 | 236 | 92.8 | 0.0 | 3434 | 2 |
|  | 10 | 0.0 | 94.6 | 98.4 | 94.5 | 0.0 | 7035 | 35 | 92.5 | 68.7 | 19033 | 333 | 94.6 | 98.0 | 1981 | 481 | 93.0 | 0.0 | 3348 | 3 |
|  | 20 | 0.0 | 94.4 | 99.6 | 94.6 | 0.0 | 14007 | 7 | 92.2 | 91.8 | 18317 | 622 | 94.4 | 99.4 | 3914 | 914 | 93.0 | 0.0 | 3253 | 2 |
| 2 / 5 | 5 | 1.1 | 94.4 | 80.4 | 94.6 | 76.4 | 7494 | 243 | 92.4 | 53.9 | 19937 | 172 | 94.7 | 79.4 | 985 | 235 | 92.6 | 0.2 | 3704 | 5 |
|  | 10 | 0.9 | 94.4 | 97.2 | 94.4 | 0.1 | 7053 | 53 | 92.9 | 81.8 | 18657 | 313 | 94.7 | 92.4 | 990 | 490 | 92.9 | 0.2 | 3200 | 16 |
|  | 20 | 1.2 | 94.4 | 94.0 | 94.5 | 87.7 | 14263 | 263 | 92.6 | 89.4 | 20531 | 406 | 94.6 | 95.4 | 966 | 966 | 92.8 | 0.3 | 3540 | 38 |
| 3 / 5 | 5 | 7.6 | 94.7 | 91.0 | 94.7 | 88.5 | 7281 | 30 | 92.6 | 90.8 | 19172 | 172 | 94.5 | 91.4 | 993 | 243 | 92.8 | 81.1 | 3693 | 167 |
|  | 10 | 5.9 | 94.8 | 94.0 | 94.4 | 8.6 | 7014 | 14 | 92.3 | 91.6 | 20293 | 296 | 94.6 | 92.3 | 988 | 488 | 92.6 | 90.2 | 3355 | 496 |
|  | 20 | 7.6 | 94.7 | 90.8 | 94.3 | 90.6 | 14092 | 92 | 92.6 | 92.6 | 18236 | 652 | 94.7 | 91.7 | 974 | 974 | 92.6 | 87.4 | 3210 | 995 |
| 3 / 7 | 5 | 0.6 | 94.3 | 33.8 | 94.6 | 98.3 | 7500 | 249 | 91.9 | 97.2 | 21289 | 246 | 94.7 | 98.4 | 995 | 245 | 92.7 | 0.0 | 3692 | 5 |
|  | 10 | 0.7 | 94.4 | 98.5 | 94.6 | 96.6 | 7135 | 135 | 92.2 | 98.2 | 20630 | 484 | 94.5 | 98.6 | 1979 | 479 | 92.7 | 2.4 | 3427 | 22 |
|  | 20 | 0.7 | 94.4 | 98.5 | 93.7 | 78.5 | 14675 | 675 | 92.3 | 98.2 | 20470 | 987 | 94.4 | 98.9 | 980 | 980 | 92.9 | 10.5 | 3259 | 43 |
| 7 / 4 | 5 | 1.5 | 94.7 | 92.0 | 94.6 | 0.6 | 7280 | 29 | 92.3 | 39.2 | 19353 | 150 | 94.8 | 91.6 | 992 | 242 | 92.8 | 0.5 | 3258 | 30 |
|  | 10 | 1.5 | 94.6 | 94.7 | 94.4 | 0.0 | 7023 | 23 | 92.0 | 47.9 | 19631 | 285 | 94.4 | 94.5 | 1979 | 479 | 93.0 | 0.3 | 3627 | 293 |
|  | 20 | 1.5 | 94.6 | 96.5 | 94.3 | 0.1 | 14000 | 0 | 92.1 | 78.6 | 21292 | 22 | 94.4 | 96.5 | 3910 | 910 | 92.8 | 84.5 | 3203 | 867 |
| 8 / 6 | 5 | 0.2 | 94.7 | 97.0 | 94.8 | 80.5 | 7470 | 219 | 92.8 | 73.3 | 19293 | 152 | 94.5 | 98.3 | 993 | 243 | 92.8 | 0.0 | 3494 | 2 |
|  | 10 | 0.2 | 94.4 | 99.5 | 94.7 | 0.0 | 7008 | 8 | 92.6 | 97.6 | 19544 | 288 | 94.4 | 99.4 | 1981 | 481 | 92.9 | 0.0 | 3571 | 1 |
|  | 20 | 0.2 | 94.7 | 99.5 | 94.4 | 0.0 | 14000 | 0 | 92.5 | 96.4 | 18946 | 597 | 94.2 | 99.5 | 3908 | 908 | 92.6 | 0.2 | 3492 | 8 |
| 9 / 2 | 5 | 0.1 | 95.0 | 92.0 | 94.4 | 93.3 | 7501 | 250 | 91.7 | 85.0 | 23573 | 151 | 94.6 | 97.3 | 988 | 238 | 93.0 | 0.0 | 3291 | 2 |
|  | 10 | 0.1 | 94.5 | 93.9 | 94.3 | 93.1 | 7500 | 500 | 92.1 | 95.1 | 22896 | 251 | 94.4 | 98.6 | 1970 | 471 | 93.1 | 0.0 | 3133 | 1 |
|  | 20 | 0.1 | 94.4 | 96.1 | 94.7 | 0.0 | 14010 | 10 | 93.1 | 98.9 | 18651 | 575 | 94.1 | 99.0 | 3906 | 906 | 93.1 | 0.0 | 3223 | 2 |

Table 12: Performance on the CIFAR-10 DLBD scenario (1-to-1) using the ResNet32 architecture, when training for the final model for 200 epochs. The S / T column lists the CIFAR-10 source and target classes. $\epsilon$ refers to the percentage of the source class which is poisoned.

| S / T | $\epsilon$ | Oracle A | ND C | ND A | SS C | SS A | SS FP | SS FN | AC C | AC A | AC FP | AC FN | ITLM C | ITLM A | ITLM FP | ITLM FN | ISPL+B C | ISPL+B A | ISPL+B FP | ISPL+B FN |
|---|---|---|---|---|---|---|---|---|---|---|---|---|---|---|---|---|---|---|---|---|
| 0 / 2 | 5 | 1.1 | 92.6 | 85.0 | 91.8 | 57.3 | 7499 | 248 | 88.8 | 0.6 | 24211 | 133 | 91.8 | 83.3 | 988 | 238 | 89.8 | 0.0 | 5752 | 31 |
| | 10 | 1.1 | 92.4 | 92.6 | 91.7 | 91.5 | 7422 | 422 | 88.8 | 78.6 | 24031 | 250 | 91.7 | 92.3 | 1969 | 469 | 89.5 | 0.0 | 5445 | 33 |
| | 20 | 1.5 | 92.5 | 94.9 | 90.4 | 64.2 | 14590 | 590 | 88.6 | 62.7 | 23686 | 455 | 91.9 | 94.2 | 3890 | 890 | 88.0 | 0.0 | 6992 | 86 |
| 1 / 3 | 5 | 0.1 | 92.7 | 98.5 | 91.7 | 5.7 | 7479 | 227 | 88.8 | 2.4 | 23903 | 125 | 91.9 | 12.7 | 988 | 238 | 89.8 | 0.0 | 5776 | 1 |
| | 10 | 0.1 | 91.6 | 35.0 | 91.1 | 3.3 | 7436 | 436 | 88.3 | 24.9 | 23891 | 254 | 92.0 | 69.2 | 988 | 488 | 89.8 | 0.0 | 5736 | 3 |
| | 20 | 0.0 | 92.1 | 98.1 | 91.5 | 0.0 | 14004 | 4 | 89.1 | 0.4 | 21145 | 70 | 91.9 | 97.3 | 3900 | 900 | 89.4 | 0.0 | 5833 | 1 |
| 2 / 5 | 5 | 1.7 | 92.2 | 62.7 | 91.2 | 43.2 | 7493 | 242 | 88.0 | 1.4 | 24001 | 128 | 92.2 | 75.7 | 994 | 244 | 88.8 | 0.1 | 6358 | 6 |
| | 10 | 1.6 | 92.5 | 92.4 | 91.9 | 89.8 | 7476 | 476 | 89.3 | 81.5 | 23827 | 275 | 91.6 | 93.2 | 1975 | 475 | 89.2 | 0.0 | 5985 | 26 |
| | 20 | 1.6 | 91.7 | 95.8 | 89.9 | 3.5 | 14349 | 349 | 88.5 | 87.2 | 21086 | 711 | 91.9 | 95.1 | 3905 | 905 | 89.8 | 0.2 | 5684 | 40 |
| 3 / 5 | 5 | 6.3 | 91.3 | 88.9 | 91.7 | 87.9 | 7466 | 215 | 88.9 | 80.0 | 23817 | 131 | 92.2 | 90.8 | 996 | 246 | 89.2 | 27.5 | 5974 | 59 |
| | 10 | 6.2 | 91.8 | 90.8 | 91.6 | 86.4 | 7348 | 348 | 88.7 | 73.1 | 21299 | 136 | 92.2 | 89.5 | 990 | 490 | 89.1 | 82.1 | 5681 | 344 |
| | 20 | 6.3 | 92.3 | 90.8 | 90.5 | 71.9 | 14090 | 90 | 89.3 | 78.5 | 20694 | 156 | 90.9 | 88.8 | 3918 | 918 | 89.8 | 86.7 | 5093 | 994 |
| 3 / 7 | 5 | 1.1 | 92.6 | 98.4 | 91.0 | 97.1 | 7452 | 201 | 86.2 | 72.8 | 23823 | 189 | 92.1 | 97.1 | 994 | 244 | 89.4 | 0.2 | 5498 | 13 |
| | 10 | 1.1 | 92.4 | 98.6 | 91.8 | 96.7 | 7315 | 315 | 88.6 | 96.2 | 23648 | 482 | 92.3 | 98.0 | 1981 | 481 | 89.8 | 0.0 | 5126 | 16 |
| | 20 | 1.1 | 92.9 | 98.6 | 90.9 | 97.0 | 14167 | 167 | 89.1 | 95.4 | 20600 | 551 | 92.4 | 98.1 | 3924 | 924 | 88.6 | 0.2 | 6642 | 26 |
| 7 / 4 | 5 | 1.7 | 92.1 | 88.5 | 91.6 | 87.5 | 7486 | 235 | 89.3 | 72.2 | 23928 | 149 | 92.4 | 92.2 | 992 | 242 | 88.8 | 0.4 | 6643 | 27 |
| | 10 | 1.7 | 92.7 | 93.9 | 91.9 | 94.2 | 7371 | 371 | 88.3 | 59.2 | 23753 | 193 | 92.1 | 96.2 | 1973 | 473 | 88.9 | 0.8 | 6478 | 32 |
| | 20 | 1.7 | 92.6 | 96.9 | 91.1 | 94.1 | 14397 | 397 | 88.2 | 47.6 | 23737 | 423 | 91.7 | 95.8 | 3904 | 904 | 88.6 | 46.9 | 6322 | 209 |
| 8 / 6 | 5 | 0.2 | 92.7 | 98.0 | 91.3 | 97.8 | 7441 | 190 | 89.1 | 88.7 | 23658 | 164 | 92.5 | 96.9 | 988 | 238 | 89.6 | 0.0 | 5991 | 0 |
| | 10 | 0.2 | 92.1 | 97.7 | 91.8 | 97.2 | 7089 | 89 | 90.2 | 95.1 | 19585 | 280 | 92.3 | 98.7 | 973 | 473 | 89.3 | 0.0 | 6007 | 2 |
| | 20 | 0.2 | 92.8 | 98.8 | 91.3 | 96.0 | 14297 | 297 | 87.2 | 64.2 | 23347 | 646 | 92.3 | 98.6 | 975 | 975 | 89.4 | 0.0 | 5691 | 3 |
| 9 / 2 | 5 | 0.1 | 92.9 | 93.2 | 91.2 | 93.2 | 7478 | 225 | 88.7 | 1.2 | 23518 | 136 | 92.1 | 94.0 | 991 | 241 | 90.3 | 0.0 | 5444 | 2 |
| | 10 | 0.1 | 92.6 | 98.6 | 91.4 | 94.7 | 7497 | 497 | 88.5 | 91.5 | 24034 | 242 | 92.5 | 97.8 | 986 | 486 | 88.1 | 0.0 | 7220 | 2 |
| | 20 | 0.1 | 92.6 | 97.4 | 90.5 | 0.0 | 14011 | 11 | 90.6 | 97.7 | 18658 | 578 | 91.9 | 99.2 | 3881 | 811 | 89.6 | 0.0 | 5742 | 1 |

### D.2.5 SENSITIVITY OF ISPL TO HYPERPARAMETERS

We next conduct some ablation studies to better understand the effects of the two main hyperparameters in Algorithm 1: the expansion factor $\alpha$ and the subset size $\beta$. Computationally, larger $\beta$ means fewer components and fewer outer iterations of ISPL (and is thus more efficient); in our main experiments, we use $\beta = 1/8$ and run ISPL 8 times in sequence to generate 8 components. Additionally, when estimating the self-expansion error, a larger $\alpha$ leads to lower variance estimates of the expansion error, but less effective measures of generalization; hence, we should prefer taking $\alpha$ as large as possible to produce better estimates (until memorization effects take over). Our experiments use both $\alpha = 1/2$ (for GTSRB, due to the imbalanced class sizes) and $\alpha = 1/4$ (for CIFAR-10).

Tables 13 and 14 present our results from the ablation studies, conducted on a subset of the CIFAR-10 DLBD and CLBD subsets. We do not include the self-training step after ISPL as in the main experiments, in order to better highlight the trends. In particular, without self-training, we expect both lower clean accuracies and higher targeted misclassification rates due to not recovering all the compatible clean data.

Our main finding is that our method is quite robust to both the expansion factor $\alpha$ and subset size $\beta$. In particular, $\beta \in [1/8, 1/16]$ and $\alpha \in [1, 1/4]$ all yield strong defenses with less than 10% targeted misclassification. As expected, increasing $\beta$ leads to a stronger defense but lower clean accuracies. In particular, $\beta = 1$ (which identifies only a single subset consisting of 96% of the original dataset) fails in all settings of $\alpha$; this is the setting for which **ISPL+B** is conceptually the most similar to the **ITLM** defense, which offers a possibly explanation for **ITLM**'s poor performance in our experiments. Additionally, $\alpha = 1$ avoids memorization, mainly because we do not train until convergence within the ISPL loop. Finally, we note that $\alpha = 1/2$ appears to achieve a modest optimum, which is particularly noticeable when paired with smaller $\beta$.

Table 13: Average clean accuracies after running ISPL on three scenarios of CIFAR-10 DLBD and CLBD using the PreActResNet18 architecture, with various settings of $\alpha$ and $\beta$.

| $\alpha$ | $\beta$ | | | | | average |
|---|---|---|---|---|---|---|
| | 1 | 1/2 | 1/4 | 1/8 | 1/16 | |
| 1 | 92.55 | 91.65 | 90.37 | 87.92 | 85.41 | 89.58 |
| 1/2 | 92.63 | 91.85 | 90.25 | 87.28 | 83.54 | 89.11 |
| 1/4 | 92.64 | 91.93 | 89.92 | 86.52 | 83.01 | 88.80 |
| 1/8 | 92.70 | 91.69 | 89.21 | 86.56 | 83.83 | 88.80 |
| 1/16 | 92.58 | 91.63 | 89.63 | 86.98 | 85.15 | 89.20 |
| average | 92.62 | 91.75 | 89.88 | 87.05 | 84.19 | |

Table 14: Average targeted misclassification rates after running ISPL on three scenarios of CIFAR-10 DLBD and CLBD using the PreActResNet18 architecture, with various settings of $\alpha$ and $\beta$.

| $\alpha$ | $\beta$ | | | | | average |
|---|---|---|---|---|---|---|
| | 1 | 1/2 | 1/4 | 1/8 | 1/16 | |
| 1 | 29.21 | 15.38 | 12.06 | 5.88 | 7.43 | 13.99 |
| 1/2 | 30.52 | 15.74 | 12.26 | 3.54 | 4.18 | 13.25 |
| 1/4 | 39.15 | 15.12 | 11.10 | 7.07 | 9.59 | 16.41 |
| 1/8 | 36.54 | 21.99 | 12.71 | 10.96 | 10.78 | 18.60 |
| 1/16 | 37.15 | 26.31 | 12.60 | 11.91 | 11.67 | 19.93 |
| average | 34.51 | 18.91 | 12.15 | 7.87 | 8.73 | |

# E ADDITIONAL RELATED WORK

**Deep Clustering.** The literature on clustering is vast, and we refer the reader to Xu & Tian (2015) for a survey of classical techniques. Classical techniques often do not scale to large, high-dimensional datasets, so recent years have seen increasing development in techniques for deep clustering. Such techniques generally use deep learning to embed some high-dimensional input data into a low-dimensional space (such as the latent space of a generative adversarial network (Mukherjee et al., 2019) or variational autoencoder (Yang et al., 2019)), then perform a classical clustering on the embeddings (Zhou et al., 2022). The structure revealed by the clustering is thus tied to the objective used to learn the low-dimensional embedding, e.g., the reconstruction loss. In contrast, our clustering mechanism is based on an interaction between the dataset and the training process, and therefore produces clusters based on the incompatibility of subsets with respect to a specific training objective (and can also be applied to any class of learned model).

**Compatibility.** Our separation results in the context of data poisoning can be viewed as clustering by exploiting weak supervision in the form of (possibly poisoned) class labels. Balcan et al. (2005) introduce a method called "co-training", and show that under a similar compatibility property, learners fit independently to two different "views" of the data can supervise each other to improve the joint performance. However, they (and similar works) assume access to a small set of trusted labels not present in our setting and do not consider the case of malicious training data.

**Boosting.** To identify which components belong together and recover the primary distribution, we use a method that can be interpreted as a basic form of boosting, in which an ensemble of weak learners is aggregated into a stronger learner. For instance, the seminal boosting algorithm AdaBoost (Freund et al., 1996) adaptively reweights the training set so that subsequent weak learners focus on samples of poor performance. The use of boosting methods for clustering has also been explored previously (Frossyniotis et al., 2004; Zhuowen Tu, 2005; Liu et al., 2007); as far as we know, we are the first to use our incompatibility objective to cluster data.

**Self-Paced Learning.** Self-Paced learning (SPL) (Kumar et al., 2010) is a type of curriculum learning (CL) (Bengio et al., 2009) that dynamically creates a curriculum by including samples of increasing difficulty based on the losses of the partially trained model. Prior works generally apply SPL to improve convergence on noisy datasets (Meng et al., 2016; Jiang et al., 2018; Zhang et al., 2020). In contrast, ISPL *discards* progressively *easier* samples to identify a subset with good expansion; to the best of our knowledge we are also the first to apply CL ideas to defend against backdoor attacks.

