# OpenReview forum: "Incompatibility Clustering as a Defense Against Backdoor Poisoning Attacks"
_ICLR.cc/2023/Conference — ICLR 2023 poster_

### Official Review · Reviewer_Sqdp · 2022-10-12

**Confidence:** 4
**Correctness:** 3
**Technical Novelty And Significance:** 3
**Empirical Novelty And Significance:** 3
**Recommendation:** 6

**Clarity, Quality, Novelty And Reproducibility:**

-Figures are in general much easier to read than tables, but Figure 2 was very difficult for me to read, since the y-axis labelings are very coarse, and I can’t tell if it is log-scale or linear if there are only two labeled tick marks per subfigure.
-The captions are not descriptive enough to understand the figures without trying to find where they are referenced in the text.  For example, in Table 1, I can’t even understand what any of the entries mean without looking at the text.  It just says they indicate “performance”.  The caption must indicate what “performance” means.  I do understand that this information is contained in Section 5.2, but it would be good to include it in the caption as well.
-Please add error bars.  I might be missing something but some of the experiments seem like they were not run very many times and therefore the statistical significance may be small.
-I appreciate the comparison to other defenses.  It should be pointed out that the defenses you compare to are all several years old and stronger defenses exist right now.

**Strength And Weaknesses:**

I really like the concept of this paper, the writing is clear, and the experimental evaluations are solid.  I especially appreciate that you consider an adaptive attacker.

I would make a suggestion though.  You might consider trying gradient-based backdoor attacks such as Sleeper Agent since (1) it may have very different properties than the types of poisons you already consider which contain patches or 8x8 watermarks in the poisons, and (2) you may be able to use it for a stronger adaptive attack.


**Summary Of The Paper:**

This paper notices that poisoned and clean "domains" do not help classify one another.  Therefore, identifying the subset of the data that does not help classify the rest can defend against poisoning attacks.  The paper also proposes a computationally feasible approach to this defense strategy and tries it on several datasets and compares its effectiveness against three existing defenses.

**Summary Of The Review:**

In general, the concept and execution are strong.  Several problems exist in the writing and some additional experiments and error bars should be added.  Nonetheless, I would tend to accept this paper.

---

> ### Author Response · Authors · 2022-11-17
> **Reply (1/1)**
>
> Thank you for the detailed review and helpful suggestions! We have been working hard to incorporate Sleeper Agent as a baseline as suggested, and are happy to share some preliminary results below (also included in the revised draft). Please let us know if there are any additional questions we can address
>
> > I really like the concept of this paper, the writing is clear, and the experimental evaluations are solid. I especially appreciate that you consider an adaptive attacker.
>
> Thanks very much for the kind comments!
>
> > I would make a suggestion though. You might consider trying gradient-based backdoor attacks such as Sleeper Agent since (1) it may have very different properties than the types of poisons you already consider which contain patches or 8x8 watermarks in the poisons, and (2) you may be able to use it for a stronger adaptive attack.
>
> Thanks for the suggestion! We have included some preliminary results as Section D.2.1 in the revised draft. A lightly edited excerpt is reproduced below. All results in this section are the mean over 24 runs, and the paper includes the standard error (removed here for clarity).
>
> We first used the official Sleeper Agent repository released by the authors to generate 24 poisoned datasets with the default settings for CIFAR-10 using different random seeds, targeting a ResNet32 as the victim architecture. We observed an attack success rate (ASR) of 49.1% over the generated datasets, training a ResNet32 model with the attacker's original training hyperparameters.
>
> We then transferred the poisoned dataset to the setting of our defense (training a ResNet32, using the same training and defense hyperparameters as in our main experiments for ResNet32 on CIFAR-10). Our defense delivers an ASR of 4.3%, compared to an oracle, which achieves an ASR of 3.0%. However, we also observed that the attack only achieves a success rate of 4.5% on an *undefended* model in our setting. We attribute this anomaly (between the ASR for an undefended model in our setting vs. the ASR reported by the attacker) to the difference in training hyperparameters, despite using the same victim architecture. In all three cases (defended, undefended, and oracle) the clean accuracy is consistent between 89-90%.
>
> We next evaluated the effectiveness of our defense when retraining with the attacker's training setup. Specifically, we took the poisoned training set generated by the attacker, then ran our defense using our training setup to output a new training set with the detected poisoned examples removed. We then used the attacker's training setup to train a new model on our new training set, and evaluated this new model on the poisoned test set.
>
> Note that in our threat model, the training algorithm $\mathcal{A}$ is fixed, and our defense is designed around the interaction between the data and $\mathcal{A}$. The above scenario is thus explicitly not the intended use case for our algorithm. Nevertheless, our defense still achieves an ASR of 14.1%, with a clean accuracy of 87.5%, compared to the original ASR of 49.1%. An oracle achieves an ASR of 2.4%.
>
> These results thus indicate that our defense successfully defends against the Sleeper Agent attack in our threat model (where the defense is integrated in the intended training pipeline), while also remaining robust to an evaluation under the attacker's original training setup.
>
> > -Figures are in general much easier to read than tables, but Figure 2 was very difficult for me to read, since the y-axis labelings are very coarse, and I can’t tell if it is log-scale or linear if there are only two labeled tick marks per subfigure. -The captions are not descriptive enough to understand the figures without trying to find where they are referenced in the text. For example, in Table 1, I can’t even understand what any of the entries mean without looking at the text. It just says they indicate “performance”. The caption must indicate what “performance” means. I do understand that this information is contained in Section 5.2, but it would be good to include it in the caption as well. -Please add error bars. I might be missing something but some of the experiments seem like they were not run very many times and therefore the statistical significance may be small.
>
> This is very helpful feedback, thanks! Hopefully our revision has addressed all these concerns. In particular,
> - Rewrote the caption of Table 1 (summary of our performance) to provide more details.
> - Rewrote the caption of Figure 2 (results from compatibility experiments) to provide more details, and clarified that the scaling is linear.
> - We have changed the way Figure 3 (comparison with baseline defenses) is presented and added error bars.

---

### Official Review · Reviewer_F4Dd · 2022-10-23

**Confidence:** 4
**Correctness:** 4
**Technical Novelty And Significance:** 3
**Empirical Novelty And Significance:** 3
**Recommendation:** 6

**Clarity, Quality, Novelty And Reproducibility:**

The proposed idea is intriguing, and the paper is well written. It appears to be reproducible for the CIFAR-10 and GTSRB datasets.


**Strength And Weaknesses:**

Strength:
- Addressing the backdoor defense is an important and timely topic right now.
- The paper is overall well written.
- The proposed incompatibility property is novel, and experimental results validate its effectiveness in defending against backdoor attacks.

Weaknesses:
- This work considers the case where the defenders have complete access to the poisoned training set. The CIFAR-10 and GTSRB datasets used in the experiments are both low-resolution, allowing the defenders to easily examine the entire dataset to find the poisoned samples. In other words, the two toy datasets used are far too simple. The successful implementation of the incompatibility property on these two datasets does not imply that the proposed incompatibility property will work well for more difficult high-resolution real-world datasets (e.g., 224 by 224 resolution images with 5 by 5 resolution trigger patterns). It would be great if the authors could demonstrate the applicability of the proposed incompatibility property to these difficult cases.
- When the defenders have no knowledge of the ratio of poisoning samples, will the proposed partition solution still be useful?


**Summary Of The Paper:**

The backdoor attack has recently received increased attention from the community. The incompatibility property is proposed in this paper in terms of the interaction of clean and poisoned data with the training algorithm, specifically that including poisoned data in the training dataset does not improve model accuracy on clean data and vice versa. The results of the experiments show that prior dirty-label and clean-label backdoor attacks in the literature produce poisoned datasets with behavior consistent with the incompatibility property. Based on this motivation, the authors propose to partition the original dataset into disjoint clean and poisoned components. Experiment results on the CIFAR-10 and GTSRB datasets show that the proposed defense method can reduce attack success rates to less than 1% in the majority of scenarios with negligible accuracy drop.

=========
Because the authors' response partially addressed my concern, I have raised my rating to 6.

**Summary Of The Review:**

Although this work has some merit, I am concerned about its applicability to more difficult real-world datasets with high-resolution images.

---

> ### Author Response · Authors · 2022-11-09
> **Reply (1/1)**
>
> Thanks very much for the supportive comments! We hope that our revision (and additional experimental results) have addressed the reviewer's main concerns. Please let us know if there are any other questions, or would like additional experiments.
>
> > This work considers the case where the defenders have complete access to the poisoned training set. The CIFAR-10 and GTSRB datasets used in the experiments are both low-resolution, allowing the defenders to easily examine the entire dataset to find the poisoned samples. In other words, the two toy datasets used are far too simple. The successful implementation of the incompatibility property on these two datasets does not imply that the proposed incompatibility property will work well for more difficult high-resolution real-world datasets (e.g., 224 by 224 resolution images with 5 by 5 resolution trigger patterns). It would be great if the authors could demonstrate the applicability of the proposed incompatibility property to these difficult cases.
>
> We have added some end-to-end experiments for the Imagenette dataset, which is a subset of ImageNet. The new results are covered in the revised version of Appendix D.2.
>
> Imagenette contains 9,469 training and 3,925 test images over 10 classes. We took the 360x360 resolution version of Imagenette, downsampled it to 224x224, then used 5x5 patches as the trigger. Examples of the clean and poisoned images are given in Table 15 in the revised version of the Appendix.
>
> For the defense, we trained a ResNet56 that achieves around 91% accuracy on the clean dataset, and otherwise used the same hyperparameters as the CIFAR-10 experiments. The experiments indicate that our approach successfully defends against 21 / 24 of the Imagenette scenarios (with all the failures occurring at the most challenging level of $\epsilon=20$), and only a 2% drop in clean accuracy (91% to 89%).
>
> > When the defenders have no knowledge of the ratio of poisoning samples, will the proposed partition solution still be useful?
>
> Our partition algorithm does not require any knowledge of the ratio of poisoning samples (apart from requiring that the poisoned data is less than 50% of the total dataset). This is supported both in theory (Theorem 3.5 and Theorem 4.2 rely only on compatibility to identify the poisoned data), as well as practice (our boosting algorithm simply rejects via majority vote, rather than using a predefined ratio). We believe this is one of the strengths of our approach over several existing baselines, which do require a target ratio for filtering.

---

### Official Review · Reviewer_HYv9 · 2022-10-25

**Confidence:** 3
**Correctness:** 3
**Technical Novelty And Significance:** 4
**Empirical Novelty And Significance:** 2
**Recommendation:** 3

**Clarity, Quality, Novelty And Reproducibility:**

The work is novel, and potentially significant.  It can be difficult to reason about the training process in the presence of poisoned data, but the incompatibility property can help.

Unfortunately, as stated above, I don't find it that clear.  I have significant questions about the basic layout of the experiments.  I believe this is easily clarified by the authors, but it must be.  Of course, this damages reproducibility significantly, as well.

**Strength And Weaknesses:**

The paper addresses the important data poisoning problem.  The approach is innovative and interesting, and effective defenses to data poisoning are rare enough that even incremental improvement on this problem is important and worth reading.  Experimental results are promising, and incompatibility provides a useful way of thinking about data poisoning.

The experiments, however, are very poorly introduced.  I was unclear if these were still being run on the maximum margin classifiers from the theoretical results, or if they were being run on something more "realistic."  I believe from the appendix they were run on ResNet variants, but it is still somewhat unclear.  This is obviously a very important point to be unclear on: does the theory, understandably developed on linear classifiers, extend well to more modern, nonlinear classifiers, or not?

I am also interested in whether the incompatibility properties hold, in theory or in practice, for other types of data poisoning that allow for distinct perturbations on individual training images.  I suspect this will be every reader's first question, and so it should be addressed in the paper.  Incompatibility is much more significant if it does hold on these other types of poisoning.

**Summary Of The Paper:**

The paper addresses the need to detect backdoor poisoned data, in which a subset of training data is permuted with a watermark, resulting in some elements of a testing set being incorrectly classified.  The paper first identifies an incompatibility property, in which poisoned training data does not improve model accuracy on clean data, and vice-versa.  They show this property applies with high probability to maximum-margin classifiers whose data have been backdoor poisoned.  They then introduce an algorithm which leverages this property to identify clean data.  Finally, they introduce an algorithm which loosens many of the requirements for the above, and tractably trains a classifier on data which is likely to be clean.

**Summary Of The Review:**

This is potentially an important result.  The bones of the work is interesting, useful, and potentially provides one of the first real defenses against data poisoning I've seen.  However, there are significant unexplained aspects of the experimental setup that make it difficult to judge and impossible to reproduce.  With editing, this would be an acceptable paper; without, it is not.

---

> ### Author Response · Authors · 2022-11-09
> **Reply (1/1)**
>
> Thanks very much for the comments, which have made our presentation more clear (particularly in the experimental setup). Please let us know if our revision has left anything unclear!
>
> > The experiments, however, are very poorly introduced. I was unclear if these were still being run on the maximum margin classifiers from the theoretical results, or if they were being run on something more "realistic." I believe from the appendix they were run on ResNet variants, but it is still somewhat unclear. This is obviously a very important point to be unclear on: does the theory, understandably developed on linear classifiers, extend well to more modern, nonlinear classifiers, or not?
>
> We sincerely apologize for the confusion.
>
> We have clarified this in our revision--all the experimental results are run with modern ResNet variants, as correctly inferred by the reviewer.
>
> > I am also interested in whether the incompatibility properties hold, in theory or in practice, for other types of data poisoning that allow for distinct perturbations on individual training images. I suspect this will be every reader's first question, and so it should be addressed in the paper. Incompatibility is much more significant if it does hold on these other types of poisoning.
>
> The results in the paper indicate that the incompatibility properties hold in practice for the three kinds of CLBD attacks ($\ell_\infty$, $\ell_2$ and GAN) on CIFAR-10, all of which use distinct perturbations for each individual training image. We have clarified that the perturbation is performed independently for each training image when introducing the CLBD attack in Section 5: “The final strategy, clean label backdoor… ***first strongly perturbs each image of the source class independently***… then inserts several patches (without changing the label)”. The fourth paragraph in Appendix B (“Dataset Construction Details”) has a more complete description of how the CLBD datasets were created.
>
> Our experimental results for the CLBD attack include both
> - (Section 5.1, and Figures 5-7 in Appendix D.1) targeted experiments to measure the incompatibility between clean and poisoned data from the CLBD attack, and
> - (Section 5.2) end-to-end experiments using our approach to defend against the CLBD attack (41 / 45 scenarios successfully defended).
>
> > The work is novel, and potentially significant. It can be difficult to reason about the training process in the presence of poisoned data, but the incompatibility property can help. Unfortunately, as stated above, I don't find it that clear. I have significant questions about the basic layout of the experiments. I believe this is easily clarified by the authors, but it must be. Of course, this damages reproducibility significantly, as well.
>
> We hope our clarifications have cleared up these questions. We note that the appendices contain a more complete picture of our experimental set up, and will also release the code to reproduce all the main experiments (namely, Table 1).
>
> > This is potentially an important result. The bones of the work is interesting, useful, and potentially provides one of the first real defenses against data poisoning I've seen. However, there are significant unexplained aspects of the experimental setup that make it difficult to judge and impossible to reproduce. With editing, this would be an acceptable paper; without, it is not.
>
> Thanks very much for the supportive comments! We apologize for the confusion caused by our initial draft, and hope that the reviewer can take a second look with the revisions.

---

### Official Review · Reviewer_ZDJY · 2022-11-01

**Confidence:** 3
**Clarity, Quality, Novelty And Reproducibility:** The clarify, quality, novelty and rep…
**Correctness:** 3
**Technical Novelty And Significance:** 3
**Empirical Novelty And Significance:** 3
**Recommendation:** 6

**Strength And Weaknesses:**

**Strengths**

1. The authors identified an interesting universal property intrinsic to the backdoor poisoning attack, which inspires the design of defense algorithms.
2. The authors conducted extensive evaluations and demonstrated effectiveness of their approach.

**Weaknesses**

I do not have major complaints about the proposed method. Below are a few comments which I hope can help improve the paper or questions to assist my understanding.

1. A highly-relevant related work [1] is missing. This submission shares a similar flavor with [1]. Both works are developed based on the intuition that clean samples will not help the performance on poisoned samples, and both works iteratively refine the subsets. I recommend the authors to add [1] in Sec 6 related work and discuss the similarities and differences.
2. The approach proposed in [1] requires access to a trained poisoned model as well as a misclassification event during testing time (their setup is forensics); your work does not assume access to these. It is pretty normal to assume the access to a trained poisoned model ([2] uses this assumption as well; see their Sec. 6). Do you think this knowledge can be leveraged to improve your approach in any principled way?
3. I understand the goal of learning a parameter $\tilde \theta$ such that it is as close as possible to $\theta^\star=\mathcal{A}(C)$ which is a model obtained on clean data only (Sec 2.2). But I have a few following thoughts: 1. How close is it to the model obtained on the original dataset $D=C \cup A$? This can be answered through experiments (and I'd appreciate it if the authors can show some results.) 2. If the gap is large in the first step (e.g., when the size of $A$ is almost the same as that of $C$, is it possible to rectify the poisoned set $P$ to its original form $A$ in some way? This could be an interesting direction for future work.
4. The definition of compatibility Def 3.2 is not symmetric. Could the authors elaborate more on this point?
5. For the Incompatibility property in Def 3.3, I'm curious how well is it satisfied in the practical attacks, and thus have the two following questions: a) Can we propose a proxy to evaluate this? b) If an attack doesn't satisfy it very well, what's the reason, and how significant it will degrade the defense?

Experiments wise:

6. Issues with Fig. 2 and the descriptions: a) left column, 2nd subfigure from the top: 0 is above 10 in the y-axis; this looks incorrect to me. b) right column, I do not understand the "source" and "non-source" for CLBD. There is no "source" but only "target" in CLBD. I find the interpretations of the subfigure difficult to understand. c) The last row shows absolute change in accuracy on poisoned data; I would love to see the absolute accuracy on poisoned data as well to understand how effective the attack is.
7. In Sec. 5.2, the authors compared to "training on the original clean samples". Does it refer to training on $C$ or $D$? (This is related to point 3.)
8. In Sec. 5.2 "Comparison with Existing Approaches", the authors said they compared with "3 existing defenses". I would recommend the authors to add the names or at least references for the three methods in this sentence.

**Minors**

* In Eq. (1), $A_{emp}$ should be $R_{emp}$.
* Typos
  * "as an absolute different in accuracy" --> "difference"



**References**

[1] Shan, Shawn, et al. "Poison Forensics: Traceback of Data Poisoning Attacks in Neural Networks." 31st USENIX Security Symposium (USENIX Security 22). 2022.

[2] Cui, Ganqu, et al. "A Unified Evaluation of Textual Backdoor Learning: Frameworks and Benchmarks." Thirty-sixth Conference on Neural Information Processing Systems Datasets and Benchmarks Track.



**Summary Of The Paper:**

This paper identifies an incompatibility property of the interaction of clean and poisoned data, i.e., involving poisoned data does not improve model performance on clean data and vice versa. The authors then leverage this property to develop a detection algorithm for finding the clean samples among the poisoned dataset. It is achieved by first separating the dataset into multiple subsets of either all clean data or all poisoned data, and then identifying the clean data among these subsets through a voting algorithm. The proposed approach ISPL+B demonstrates great performance against various types of backdoor attacks.

**Summary Of The Review:**

The problem is important and the solution is novel. The paper quality is good. I only have a few comments and questions. I'm leaning to acceptance of the paper.

---

> ### Author Response · Authors · 2022-11-09
> **Reply (1/2)**
>
> We would like to thank the reviewer for the helpful comments! Please let us know if anything remains unclear after reading our response.
>
> > A highly-relevant related work [1] is missing. This submission shares a similar flavor with [1]. Both works are developed based on the intuition that clean samples will not help the performance on poisoned samples, and both works iteratively refine the subsets. I recommend the authors to add [1] in Sec 6 related work and discuss the similarities and differences.
>
> Thanks for bringing this work to our attention! As suggested, we have added a discussion of this work to Section 6 in our revision. For clarity, we also provide a more extensive discussion below.
>
> We believe the reviewer has identified the main similarities, namely, that we both use an iterative procedure, and also rely on an intuition that training on clean samples does not help the performance on poisoned samples (note that we also require the reverse assumption as well, namely, that training on poisoned samples does not help the performance on clean samples). Apart from the setting (forensics vs. defending), the major difference is in how we formalize and exploit this intuition.
>
> Briefly, [1] relies on k-means clustering (based on the gradient of the parameters in the final linear layer, evaluated at input points, taken with respect to the uniform distribution over the labels) to produce a clustering, and only then do they leverage the assumption to **identify** which cluster contains poisoned data. As far as we can tell, the clustering mechanism (Equation 2 in Section 4.2 in [1] under “Data Mapping”) is more directly motivated by “unlearning”, rather than this common intuition.
>
> Conversely, in our case, we formalize the intuition in Definition 3.3, then leverage this as the theoretical basis of the **clustering** step in our defense (Theorem 3.5). Then we use a boosting algorithm to identify the clusters containing the clean data--the only additional assumption is that the poisoned data comprises less than 50% of the dataset. That one can essentially leverage our formal notion of incompatibility as the sole workhorse of a (theoretically justified) defense is a contribution of our work.
>
> Finally, for completeness, we emphasize that [1] performs forensics (so that they begin analysis with a known misclassification event), whereas our defense operates in a more difficult black-box setting.
>
> [1] Shan, Shawn, et al. "Poison Forensics: Traceback of Data Poisoning Attacks in Neural Networks." 31st USENIX Security Symposium (USENIX Security 22). 2022.
>
> > The approach proposed in [1] requires access to a trained poisoned model as well as a misclassification event during testing time (their setup is forensics); your work does not assume access to these. It is pretty normal to assume the access to a trained poisoned model ([2] uses this assumption as well; see their Sec. 6). Do you think this knowledge can be leveraged to improve your approach in any principled way?
>
> In our case, we could also train a model on the poisoned training set $C \cup P$; however, our threat model assumes that the model should achieve low loss on the training set, and hence, there is not much information to be gained. ([1] gets around this by making an implicit assumption that the gradients of the trained model still contain information, but we do not see any obvious way to either justify this claim, or use it in our defense.)
>
> On the other hand, having access to a known misclassification event (x, y) would allow us to perform identification more directly--rather than boosting, we can just reject any weak learners whose outputs agree with the known misclassification event.

---

> > ### Author Response · Authors · 2022-11-09
> > **Reply (2/2)**
> >
> > > I understand the goal of learning a parameter such that it is as close as possible to which is a model obtained on clean data only (Sec 2.2). But I have a few following thoughts: 1. How close is it to the model obtained on the original dataset ? This can be answered through experiments (and I'd appreciate it if the authors can show some results.)
> >
> > The results in Section D.2 include both “Oracle” (training on just $C$) and “No defense” (training on $C \cup P$) for all the experiments used in the main text (CIFAR-10 DLBD, Watermark, CLBD, and GTSRB DLBD, Watermark). In general the performance of the Oracle model appears to be slightly higher (by about 0.2% on both CIFAR-10 and GTSRB).
> >
> > We can also compare the Oracle performance with a model trained on the original dataset (ie., the entire clean CIFAR-10 or GTSRB training set, which is referred to as $D$ in the paper). In this case, the performances are pretty much identical (93% on CIFAR-10, 94.5% on GTSRB).
> >
> > If the reviewer has additional experiments in mind, we would be happy to try and run them.
> >
> > > 2. If the gap is large in the first step (e.g., when the size of is almost the same as that of , is it possible to rectify the poisoned set to its original form in some way? This could be an interesting direction for future work.
> >
> > Rectifying the poisoned data would likely require different techniques, and is certainly an interesting direction for future work.
> >
> > > The definition of compatibility Def 3.2 is not symmetric. Could the authors elaborate more on this point?
> >
> > The two equations in Definition 3.2 are for compatibility and incompatibility, where we have just reversed the direction of the inequality. (If this does not answer the question, perhaps the reviewer could clarify?)
> >
> > > For the Incompatibility property in Def 3.3, I'm curious how well is it satisfied in the practical attacks, and thus have the two following questions: a) Can we propose a proxy to evaluate this?
> >
> > In Figure 2, we present some results evaluating the incompatibility between randomly selected subsets of clean and poisoned data, taken from the DLBD and CLBD attacks on CIFAR-10. Appendix D.1 also contains the complete results of this experiment (for the Watermark attack on CIFAR-10, as well as both DLBD and Watermark attacks on GTSRB), as well as an extensive discussion of the results. All our results indicate that the key inequality is empirically satisfied by existing attacks.
> >
> > > b) If an attack doesn't satisfy it very well, what's the reason, and how significant it will degrade the defense?
> >
> > For attacks which violate the incompatibility property, our defense does not provide any guarantees that it will successfully remove the poisoned data.
> >
> > > Issues with Fig. 2 and the descriptions: a) left column, 2nd subfigure from the top: 0 is above 10 in the y-axis; this looks incorrect to me.
> >
> > Thanks! This has been fixed. The 10 should have been -10.
> >
> > > b) right column, I do not understand the "source" and "non-source" for CLBD. There is no "source" but only "target" in CLBD. I find the interpretations of the subfigure difficult to understand.
> >
> > We have clarified our terminology in the revision--please let us know if it is still unclear. For CLBD, we treat the “source” and “target” classes as the same class. For instance, if the target class of the CLBD attack is airplane, then the second subfigure (“Source”) plots only the performance on airplanes, and the third subfigure (“Non-source”) plots the performance on the non-airplane classes.
> >
> > > c) The last row shows absolute change in accuracy on poisoned data; I would love to see the absolute accuracy on poisoned data as well to understand how effective the attack is.
> >
> > Since the baseline (first column) is the accuracy on poisoned data when there is no poison in the training set, the baseline accuracy on poisoned data is (close to) 0, and so the absolute accuracy is very close to what you see as the change in accuracy as well.
> >
> > > In Sec. 5.2, the authors compared to "training on the original clean samples". Does it refer to training on C or D? (This is related to point 3.)
> >
> > We are referring to training on $D$. Thanks for pointing this out--we have clarified this in the revision!
> >
> > > In Sec. 5.2 "Comparison with Existing Approaches", the authors said they compared with "3 existing defenses". I would recommend the authors to add the names or at least references for the three methods in this sentence.
> >
> > Done! Thanks for the suggestion.

---

### Author Response · Authors · 2022-11-09
**Revisions uploaded**

We would first like to thank all the reviewers for the excellent feedback. Based on the reviews, we have uploaded a revised version of the paper to address the main comments. The major changes are highlighted in the paper, and detailed below:

Section 5 (Experimental Evaluation)
- Rewrote the caption of Table 1 (summary of our performance) to provide more details
- Rewrote the caption of Figure 2 (results from compatibility experiments) to provide more details
- We have changed the way Figure 3 (comparison with baseline defenses) is presented and added error bars.

Section 6 (Related Work)
- Added a discussion of related work [1], and moved the sections on Compatibility and Self-Paced Learning to the appendix.

Appendix D.2 (Additional Experimental Results, Defense Evaluation)
- Added results for a high-resolution dataset (Figure 15 and Table 9), which addresses a concern of Reviewer F4Dd.

[1] Shan, Shawn, et al. "Poison Forensics: Traceback of Data Poisoning Attacks in Neural Networks." 31st USENIX Security Symposium (USENIX Security 22). 2022.

---

> ### Author Response · Authors · 2022-11-17
> **Additionall Revisions**
>
> We have uploaded a newly revised document with the following major changes:
>
> Appendix D.2 (Additional Experimental Results, Defense Evaluation)
> - Added results for the Sleeper Agent attack, which addresses a suggestion by Reviewer Sqdp.
> - Moved the results for a high-resolution dataset to Appendix D.2.2.

---

### Decision · Program_Chairs · 2023-01-20

**Decision:**

Accept: poster

**Justification For Why Not Higher Score:**

The experimental part is ok but not very strong for this paper. Also, clarity can be improved.

**Justification For Why Not Lower Score:**

The proposed idea is novel and interesting.

**Metareview: Summary, Strengths And Weaknesses:**

The paper formally defines "incompatibility property" of the interaction between clean and poisoned data and shows how to use this idea to detect poisoned data. Although all the reviewers agree that the paper is novel and interesting, there were several questions raised by the reviewers in the initial phase including some clarification questions (e.g., what kind of models are used in the experiments, whether the method can work for higher resolution, and what kind of knowledge about poisoned data is required by the algorithm). This paper initially had a low average score (5) and was not listed as borderline (so no zoom meeting was scheduled).

In the rebuttal phase, most of the important concerns are addressed, and the paper ends up getting 3 weak accepts and 1 reject in the very late stage. The AC carefully checked the reviews and rebuttals and found that most of the concerns are already addressed, and all the reviewers (even the rejection one) agree that the paper is interesting and novel.  The AC also agree with the reviewers that the idea of incompatibility is interesting, especially how to formulate it mathematically. As the reviewer who gave score 3 (reject) did not respond to rebuttal and several reminders sent by the AC, and the comments from the original review have been addressed, the AC decide to recommend acceptance for this paper.

**Note From Pc:**

if the above contains the word "oral" or "spotlight" please see: "oral" presentation means -> notable-top-5% and "spotlight" means -> notable-top-25%. As stated in our emails, we are disassociating presentation type from AC recommendations